# A peptide tag-specific nanobody enables high-quality labeling for dSTORM imaging

David Virant[1], Bjoern Traenkle[2], Julia Maier[2], Philipp D. Kaiser[3], Mona Bodenhöfer[3], Christian Schmees[3], Ilijana Vojnovic[1], Borbála Pisak-Lukáts[1], Ulrike Endesfelder[1] & Ulrich Rothbauer[2,3]

Dense fluorophore labeling without compromising the biological target is crucial for genuine super-resolution microscopy. Here we introduce a broadly applicable labeling strategy for fixed and living cells utilizing a short peptide tag-specific nanobody (BC2-tag/bivBC2-Nb). BC2-tagging of ectopically introduced or endogenous proteins does not interfere with the examined structures and bivBC2-Nb staining results in a close-grained fluorophore labeling with minimal linkage errors. This allowed us to perform high-quality dSTORM imaging of various targets in mammalian and yeast cells. We expect that this versatile strategy will render many more demanding cellular targets amenable to dSTORM imaging.

[1] Department of Systems and Synthetic Microbiology, Max Planck Institute for Terrestrial Microbiology and LOEWE Center for Synthetic Microbiology (SYNMIKRO), Karl-von-Frisch Strasse 16, Marburg 35043, Germany. [2] Pharmaceutical Biotechnology, Eberhard Karls University Tuebingen, Markwiesenstrasse 55, Reutlingen 72770, Germany. [3] Natural and Medical Sciences Institute at the University of Tuebingen, Markwiesenstrasse 55, Reutlingen 72770, Germany. These authors contributed equally: David Virant, Bjoern Traenkle, Ulrike Endesfelder and Ulrich Rothbauer. Correspondence and requests for materials should be addressed to U.E. (email: ulrike.endesfelder@synmikro.mpi-marburg.mpg.de) or to U.R. (email: ulrich.rothbauer@uni-tuebingen.de)

Fluorescence-based super-resolution microscopy (SRM) is becoming increasingly applied in cell biology. Single-molecule localization microscopy (SMLM) techniques, such as (direct) stochastic optical reconstruction microscopy ((d)STORM) provide outstanding spatial resolutions and have enabled unprecedented insights into the organization of sub-cellular components[1–3]. However, the quality and value of SMLM imaging can be limited due to poor photon emission or detection efficiency, low fluorophore labeling densities, linkage errors or steric hindrances[4–6]. Most current SMLM labeling approaches employ antibodies or recombinant proteins either fused to pho-toactivatable fluorescent proteins (FPs) or fluorogen-labeling enzymes, such as the Halo-, CLIP-, or SNAP-tag[7–10]. While conventional antibodies introduce significant linkage errors by displacing the fluorophore from the target, large protein/enzyme tags can affect expression, cellular localization, folding and/or function of the respective fusion protein[11–13]. Although small peptide tags, such as FLAG-, HA-, or Myc-tag[14–16] are available, those epitopes often have to be arranged in multiple arrays to recruit medium-affine binding antibodies[17] and thus do not provide dense labeling sufficient for high-quality SRM.

Instead of using antibodies, a 15-amino-acid peptide-tag can be visualized by high-affinity fluorescently labeled monomeric streptavidin[18], which, however, can be affected by the binding of endogenously biotinylated proteins. Alternatively, reversibly on- and off-binding labels in point accumulation for imaging of nanoscale topography (PAINT) microscopy allow for a con-tinuous and therefore ultra-high density readout as they are not limited by a predefined fluorophore tagging pattern[19]. Yet, this approach can only be used for distinguishable structures like membranes or DNA combined with illumination-confined arrangements, such as in surface-near or lightsheet illumina-tions[20]. The visualization of other structures by PAINT approa-ches relies on a specific labeling commonly achieved by DNA-PAINT[21, 22].

As a promising substitute for conventional antibodies, small-sized nanobodies (antibody fragments derived from heavy-chain-only camelid antibodies) coupled to organic dyes were recently introduced for SRM. Nanobodies targeting native proteins, such as components of the nuclear pore complex, tubulin, or vimentin were described for dSTORM imaging[23–25]. Despite their cap-ability to directly probe endogenous antigens, the de novo gen-eration of gene-specific nanobodies and their validation for SRM imaging purposes is cumbersome and time-consuming[26, 27], which is reflected by the fact that only a very limited number of SRM-compatible nanobodies are available by now[25]. Due to their applicability for nanoscopy of widely used FP-fusions, GFP-, and RFP-nanobodies became very popular tools for SMLM[28, 29]. However, this strategy relies on the correct expression of FP-fusions and does not cope with problems arising from mis-localization or dysfunction[12, 13, 30]. Thus, nanobodies directed against short and inert tags might prove advantageous for SRM.

Here we introduce a versatile labeling and detection strategy comprised the short and inert BC2 peptide-tag (PDRKAAVSHWQQ) and a corresponding high-affinity biva-lent nanobody (bivBC2-Nb) for high-quality dSTORM imaging. We demonstrate the benefits of our approach for close-grained fluorophore labeling with minimal linkage error of various ectopically introduced and endogenous targets in fixed and living cells.

## Results

### Development of a dSTORM suitable BC2-tag/bivBC2-Nb sys-tem. As originally described, we first labeled the BC2-Nb at accessible lysine residues by N-hydroxysuccinimide (NHS) ester

fluorophores, such as Alexa Fluor 647 (AF647)[31]. While BC2-Nb$_{AF647 (NHS)}$ is sufficient for wide-field microscopy (Fig. 1a, left panel, Supplementary Fig. 1a, b), dSTORM imaging of BC2-tagged proteins revealed a rather low-staining efficiency resulting in inferior structural labeling coverage (Fig. 1b, left panel). Thus, we analyzed the binding properties of a bivalent format of the BC2-Nb (bivBC2-Nb) (Fig. 1a, right panel). We assessed its binding kinetics by biolayer interferometry (BLI) and observed a considerably reduced dissociation rate compared to monovalent BC2-Nb (Supplementary Fig. 1c). Notably, this decrease in dis-sociation rate is not caused by simultaneous binding of the bivBC2-Nb to two BC2 epitopes as confirmed by a BLI assay using a tandem-BC2-tag of two consecutively linked BC2 epitopes (BC2-BC2-tag) (Supplementary Fig. 1d).

Nevertheless, antigen labeling using the bivBC2-Nb conjugated to NHS-ester fluorophores (bivBC2-Nb$_{AF647 (NHS)}$) did not yield the expected visual improvement of staining specificity (Fig. 1a, right panel). Considering the crystal structure[31], we designed a site-directed, enzymatic coupling strategy, which should not affect the paratope and binding properties of bivBC2-Nb. Using the Sortase A system, we linked peptides conjugated to a single-AF647 fluorophore in a defined 1:1 ratio to the C-terminus of bivBC2-Nb (bivBC2-Nb$_{AF647 (sort)}$)[32, 33]. This approach signifi-cantly improved the staining specificity by a factor of two compared to bivBC2-Nb$_{AF647 (NHS)}$ (Fig. 1a, right panel, Fig. 1c and Supplementary Fig. 1a, b, e). An exemplary dSTORM image of a HeLa cell transiently expressing vimentin$_{BC2T}$ and stained with the bivBC2-Nb$_{AF647 (sort)}$ illustrates the remarkable quality of the BC2-tag/bivBC2-Nb labeling strategy (Fig. 1b, right panel). For a better understanding bivBC2-Nb$_{AF647 (sort)}$ is referred to bivBC2-Nb$_{AF647}$ in the following.

Since the BC2-Nb was originally developed against β-catenin[34], we assessed the influence of the background staining of endogenous β-catenin on the labeling quality. To distinguish background due to general unspecific staining from additional β-catenin staining, we compared HeLa cells (not expressing any GFP epitope) stained with a GFP-targeting nanobody (GFP-Nb$_{AF647}$) to HeLa cells stained with bivBC2-Nb$_{AF647}$. Further, we performed bivBC2-Nb$_{AF647}$ staining in HeLa cells transiently expressing the non-structural, autophagosomal marker protein LC3B fused to the BC2-tag ($_{BC2T}$LC3B), which is - in the absence of autophagy- homogeneously distributed throughout the cyto-plasm. By analyzing the dSTORM data using DBSCAN cluster-ing[35], we measured a slightly increased level of 1.7 (±0.3 S.D.) nanobodies per square micrometer for bivBC2-Nb$_{AF647}$ com-pared to the unspecific background staining of 0.61 (±0.03 S.D.) GFP-Nb$_{AF647}$ per μm$^2$. However, this level is considerably lower compared to 7.2 (±1.3 S.D.) bivBC2-Nb$_{AF647}$ per μm$^2$ which we obtained for the staining of $_{BC2T}$LC3B expressing cells (Fig. 1d, Supplementary Fig. 2a). We then compared signal intensities derived from bivBC2-Nb$_{AF647}$-stained HeLa cells, which were either left untreated or incubated with CHIR99021 (CHIR) to accumulate endogenous β-catenin[34]. While immunolabeling with a β-catenin-specific antibody showed a strong enrichment in CHIR-treated cells (Supplementary Fig. 2b), dSTORM imaging revealed only a minor increase of bivBC2-Nb$_{AF647}$ localizations (Supplementary Fig. 2c, left panel). Moreover, in CHIR-treated HeLa cells transiently expressing vimentin$_{BC2T}$, the nanobody signal was almost exclusively detectable at vimentin fibers (Supplementary Fig. 2c, right panel). Overall, bivBC2-Nb$_{AF647}$ staining resulted in 36 (±2 S.D.) localizations per μm$^2$ for untreated HeLa cells, 133 (±5 S.D.) localizations per μm$^2$ for CHIR-treated HeLa cells, 2493 (±285 S.D.) localizations per μm$^2$ for HeLa-vimentin$_{BC2T}$ cells and 2490 (±456 S.D.) localizations per μm$^2$ for CHIR-treated HeLa-vimentin$_{BC2T}$ cells (Supplemen-tary Fig. 2d). From this we conclude that even if present at high

levels, the BC2-epitope of β-catenin has a negligible impact on staining of ectopically introduced antigens.

For a stoichiometric quantification of the labeling quality of the BC2-tag/bivBC2-Nb detection system, we utilized the *Escherichia coli* protein ferritin (FtnA) recently described as a homo-oligomeric protein standard of 24 subunits[36]. We expressed BC2-tagged, as well as SNAP-tagged FtnA-24mers in U2OS cells and performed dSTORM imaging on cell lysates immobilized on coverslips[36]. By measuring single-AF647 blinking events, we obtained the parameters of the corresponding log-normal

distribution ($\mu = 5.68$, $\sigma = 0.4$), which describes the probability distribution of single-molecule fluorescence intensities (Supplementary Fig. 3a). We then measured the fluorescence intensities of immobilized FtnA oligomers labeled with the BC2-tag/bivBC2-Nb or SNAP-tag system. We compared these distributions to the expected fluorescence intensity distributions of fully labeled FtnA-24mers, calculated from the single-molecule fluorescence intensity distribution and the degree of labeling of each component (Methods section). As a result, the BC2-tag/bivBC2-Nb FtnA-oligomer staining revealed a completeness of labeling of

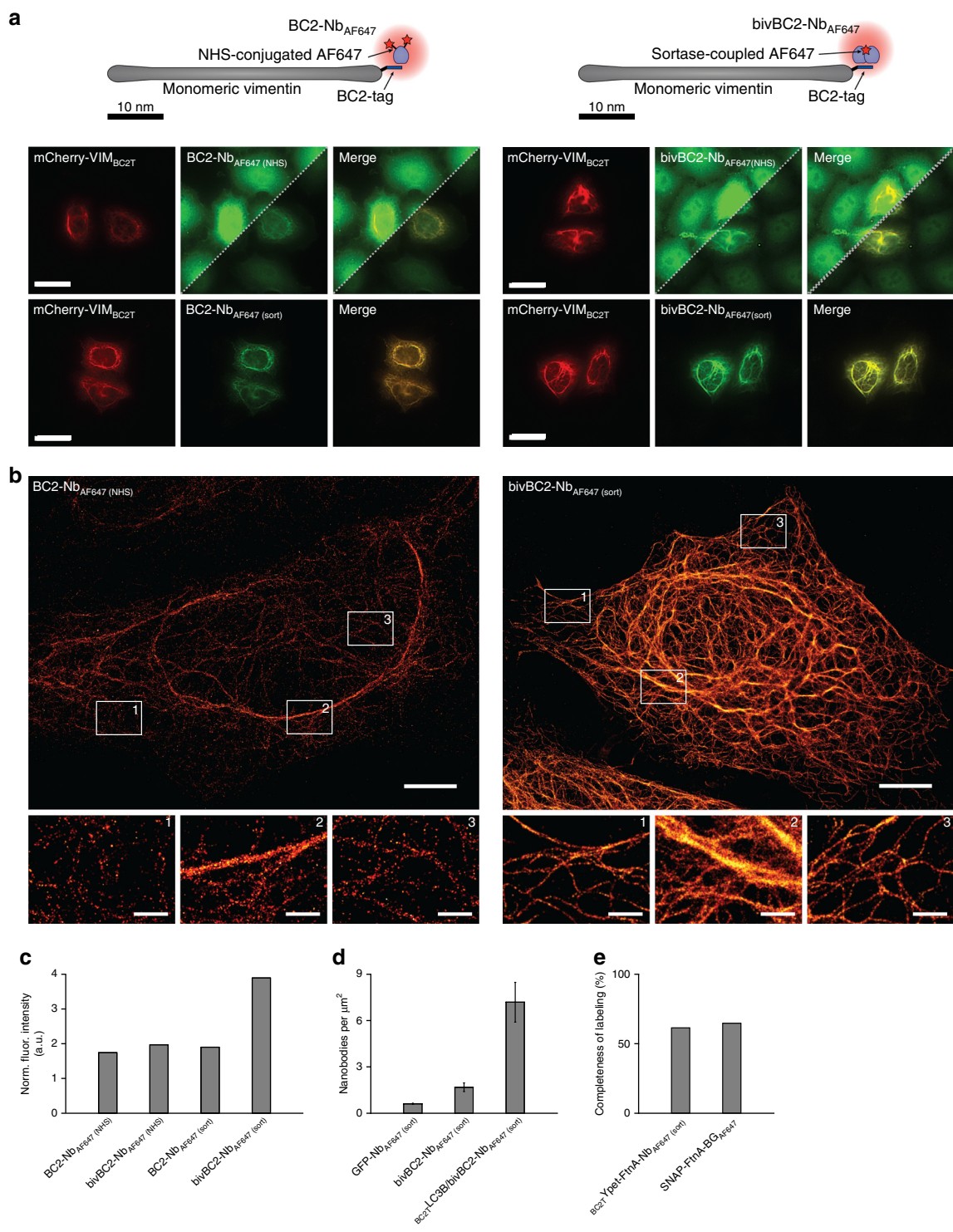

61.4% which competes with the covalent SNAP-tag FtnA-staining efficiency of 64.7% (Fig. 1e, Supplementary Fig. 3b) and outcompetes photoactivation/photoconversion efficiencies of fluorescent proteins[37]. Notably, our observation of a rather low efficiency of about 65% for the SNAP-tag labeling is in agreement with reported assessments[36, 38], and conference presentations by K. Yserentant (2017).

**Comparison of different labeling strategies for SMLM.** As genetic tagging of structural proteins like vimentin often impairs their structure and function[39, 40], we evaluated the influence of the short BC2-tag on vimentin structure formation and compared the BC2-tag/bivBC2-Nb detection system with established strategies focusing on image quality and apparent organization of the vimentin network. We performed SMLM on native vimentin in comparison to vimentin fused to photoactivatable mCherry (PAmCherry-vimentin), eGFP (GFP-vimentin), or the BC2-tag (Fig. 2a, b). For our studies, we transiently expressed the corresponding proteins for 24 h in HeLa cells followed by chemical fixation of the cells. Native vimentin was visualized with the recently described vimentin-specific nanobody bivVB6-Nb$_{AF647}$[25, 41], while PAmCherry-vimentin was mapped directly. The other constructs were labeled with the nanobodies GFP-Nb$_{AF647}$ or bivBC2-Nb$_{AF647}$, respectively. Image analysis of the vimentin network visualized by the different labeling strategies revealed considerable phenotypic differences (Fig. 2b, c; images of all cells quantitatively analyzed Supplementary Fig. 4a–d; analysis routine Supplementary Fig. 4e and Methods section). 94% of native vimentin fibers labeled with the bivVB6-Nb$_{AF647}$ showed widths lower than 150 nm. In contrast, cells with incorporated PAmCherry-vimentin were smaller and showed a high percentage (20%) of thick vimentin bundles above 150 nm width, while in GFP-vimentin expressing cells more than 96% of all detectable vimentin fibers had widths below 75 nm (Fig. 2c). Compared to the N-terminally labeled counterparts, cells expressing vimentin-PAmCherry displayed a highly similar phenotype whereas cells expressing vimentin C-terminally fused to GFP (vimentin-GFP) showed an even more severely fragmented vimentin network (Supplementary Fig. 5). Obviously, both type and position of the FP affects the formation of the vimentin network and induce altered cellular phenotypes. The various observed morphological alterations are likely caused by several mislocalization and self-oligomerization artifacts induced by the different FP moieties derived either from jelly fish (GFP) or red corals (DsRed)[40, 42]. Notably, no phenotypic changes or significant differences in the abundance of fiber widths were detected between native and BC2-tagged vimentin (96% of all fibers below 150 nm, 4% above 150 nm; Fig. 2b, c and Methods section).

We then assessed the SMLM image quality achievable by the different labeling approaches. The quality is dependent on two main factors; (i) the optical resolution dictated by the precision with which fluorescent spots can be localized, and (ii) the structural resolution determined by the labeling density (coverage) and the physical distance between fluorophore and target (linkage error). We assessed these parameters for each analyzed fiber individually. The localization precision was calculated by a Nearest Neighbor based Analysis (NeNA)[43], the labeling density was determined by the lengthwise fluorescent signal coverage along each fiber, and the linkage error by quantifying the apparent width of fibers of the smallest fiber category. For further comparison, we calculated the Fourier Image REsolution (FIRE) values[44] (see Supplementary Note 1). Since the readout of all three nanobody labeling strategies relies on the same bright fluorophore (AF647), NeNA yielded the same optical resolution statistics with a mean NeNA localization precision of about 9–12 nm. The fluorescent-protein PAmCherry has a lower photon yield and achieves an average NeNA value of 17 nm (Supplementary Fig. 6). The structural resolution as assessed by the different labeling coverage statistics revealed significant differences (Fig. 2c, Supplementary Fig. 6 and Methods section). For PAmCherry-vimentin, we observed the lowest coverage among all labeling strategies for thin fibers, and a maximum coverage of ~75% for thick fibers, which is likely due to inefficient chromophore formation and photoactivation. The low coverage of ~50% for the GFP-Nb is more likely explained by a steric hindrance in incorporating GFP-tagged molecules into the native vimentin network, which is in line with our observation of only thin fibers. The highest labeling coverage was observed for bivBC2-Nb with a coverage of ~80% for fibers below 75 nm width, and nearly full coverage of fibers exceeding a width of 150 nm. For thin fibers it exceeds the coverage obtained with the bivVB6-Nb for native vimentin, which might be due to a reduced accessibility of the native epitope within assembled vimentin filaments. To assess the impact of the size of the labeling probe on the structural resolution, we compared our bivBC2-Nb-based approach with conventional, monoclonal antibody staining (Supplementary Fig. 7a). Antibody labeling resulted in nearly complete coverage of thin vimentin fibers (>75 nm) (Supplementary Fig. 7b), and the AF647-based readout resulted in the same localization precision and optical

**Fig. 1** Comparison and characterization of BC2-nanobody (BC2-Nb) formats for wide-field and dSTORM imaging. **a** Schematic illustration of the BC2-Nb dye-conjugation strategies. Monovalent and bivalent BC2-Nbs were either conjugated with Alexa Fluor 647 (AF647) via N-hydroxysuccinimide (NHS) ester (left panel) or linked to AF647 by enzymatic sortase coupling (right panel). Wide-field imaging of chemically fixed HeLa cells expressing mCherry-vimentin$_{BC2T}$ (mCherry-VIM$_{BC2T}$) stained with modified BC2-Nbs. Monovalent versions of the BC2-Nbs (NHS- and sortase-coupled) are depicted on the left panel, corresponding bivBC2-Nbs are displayed on the right side. Stainings with NHS-conjugated nanobodies are shown in two different image contrasts, the upper half in the same brightness and contrast as the sortase-coupled nanobodies; in the lower half with an adjusted contrast. Scale bars, 25 µm. **b** Representative dSTORM images of chemically fixed HeLa cells expressing vimentin$_{BC2T}$, stained with the monomeric NHS-conjugated BC2-Nb$_{AF647}$ (NHS) (left) and the sortase-coupled bivBC2-Nb$_{AF647}$ (sort) (right). Scale bars, images 5 µm, insets 1 µm. Image reconstruction details are given in Methods section. **c** Assessment of staining quality in wide-field fluorescence imaging. Labeling of the different nanobody formats was quantified by calculating the ratio of the signal intensity of mCherry-VIM$_{BC2T}$ expressing cells to non-transfected cells (background), (BC2-Nb$_{AF647}$ (NHS): $n = 115$; bivBC2-Nb$_{AF647}$ (NHS): $n = 134$; BC2-Nb$_{AF647}$ (sort): $n = 150$; bivBC2-Nb$_{AF647}$ (sort): $n = 195$) (Methods section, Supplementary Fig. 1). **d** Assessment of bivBC2-Nb$_{AF647}$ staining of endogenous β-catenin. Bar chart summarizes measured nanobody per µm$^2$ values for untransfected chemically fixed HeLa cells stained with GFP-Nb$_{AF647}$ or bivBC2-Nb$_{AF647}$ in comparison to chemically fixed HeLa cells transiently expressing $_{BC2T}$LC3B stained with bivBC2-Nb$_{AF647}$, errors given as standard deviation (S.D.), $N = 3$ cells for each condition (Methods section, Supplementary Fig. 2). **e** Quantification of completeness of labeling for the bivBC2-Nb and SNAP-tag labeling systems using FtnA-oligomers of 24 subunits. Bar chart summarizes median values of FtnA-24mer fluorescence intensities as percentage of theoretical maxima (Methods section, Supplementary Fig. 3)

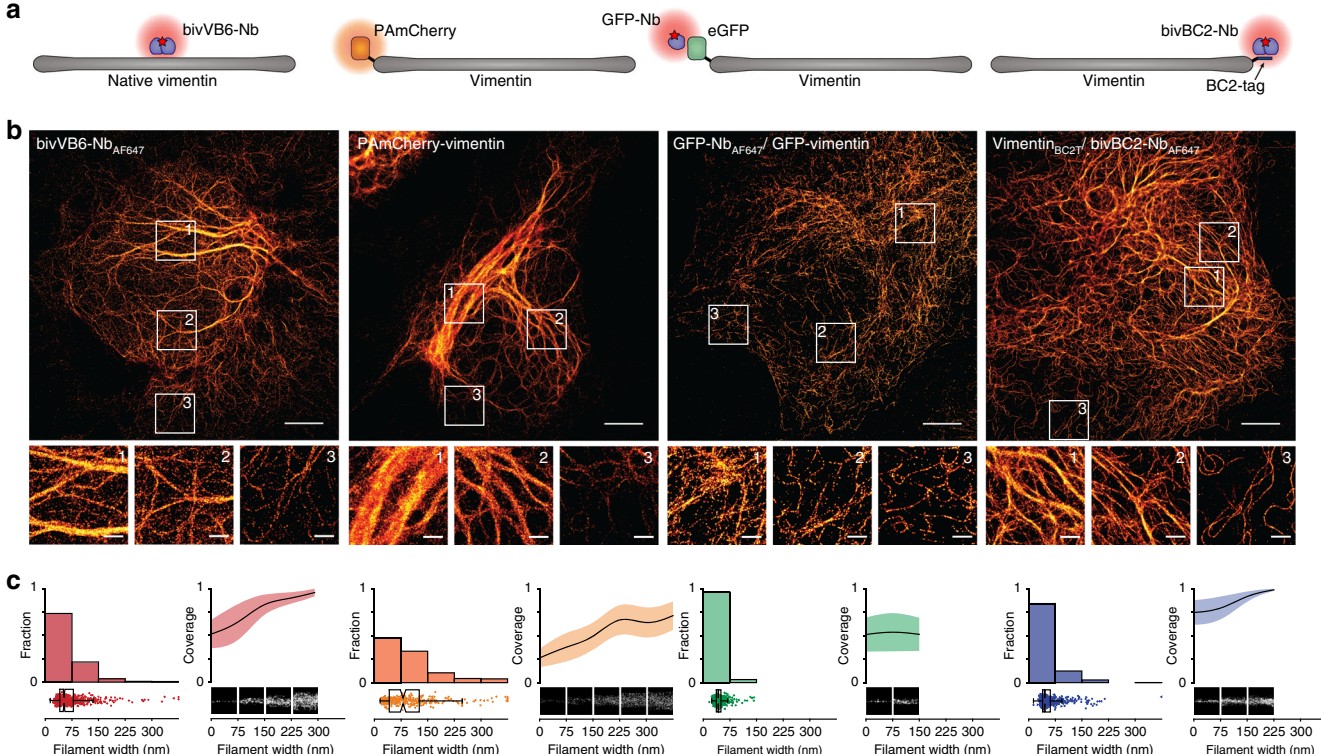

**Fig. 2** Super-resolution imaging and analysis of differently labeled vimentin constructs. **a** Schematic illustration of labeling strategies used for comparative SRM imaging of native or ectopically expressed vimentin. **b** Representative PALM/dSTORM images of chemically fixed Hela cells expressing the corresponding constructs outlined in **a** or native vimentin (left panel). Insets show magnifications of representative vimentin filaments of varying thickness (1—thick, 2—medium, 3—thin, peripheral). Scale bars, 5 µm in main images, 1 µm in insets. **c** Filament widths as histograms (left) with a bin size of 75 nm (x-axis) plotted against relative fraction (y-axis). Full data are represented underneath the histograms as box + scatter plots with the same x-axis. The box marks the three quartiles and the whiskers mark 95% of all the data. The average lengthwise fluorophore coverage was calculated for each bin and plotted (right) as mean filament width (black line) and standard deviation (colored area) against relative fraction covered by fluorophores (y-axis). Width and lengthwise fluorophore coverage were analyzed for a total of 676 (bivVB6-Nb$_{AF647}$), 295 (PAmCherry), 724 (GFP-Nb$_{AF647}$), and 620 (bivBC2-Nb$_{AF647}$) filaments, $N = 5$ cells for each condition, cells, and selected filaments are shown in Supplementary Fig. 4. Image reconstruction details are given in Methods section

resolution statistics (Supplementary Fig. 7c). Despite the high-labeling coverage, the antibody-mediated displacement of the fluorophore led to an increased linkage error. Accordingly, we measured an average width of ~55 nm for thin vimentin fibers probed with the antibody, whereas a smaller apparent width of ~40 nm was observed with bivBC2-Nb (Supplementary Fig. 7c).

**Detection of various cellular targets with the bivBC2-Nb**. Next we analyzed whether the BC2-tag/bivBC2-Nb detection system is transferable to other structural proteins. To test whether orientation of the BC2-tag affects the incorporation of the recombinant protein into endogenous structures, we transiently expressed cDNAs of mouse *TUBA1B*, human *LMNB1*, or *ACTB* either comprising the BC2-tag on the N- or the C-terminus in different cell lines followed by detection with bivBC2-Nb$_{AF647}$. As exemplarily shown for tubulin alpha-1B, C-terminal addition of the BC2-tag yielded more distinct microtubule structures compared to the N-terminally tagged version (Supplementary Fig. 8). For lamin B1, we observed no differences regarding the tag position whereas ectopically expressed β-actin is only incorporated into the actin cytoskeleton when the BC2-tag is located at the N-terminus, which is in accordance to previously tested tagging approaches[45]. dSTORM imaging of HeLa cells expressing either lamin$_{BC2T}$ or $_{BC2T}$actin, as well as U2OS cells transiently

expressing tubulin$_{BC2T}$ revealed that BC2-tagged proteins are efficiently incorporated in the corresponding structures and could be imaged at high-resolution reaching localization precisions of 9–12 nm as previously shown for vimentin$_{BC2T}$ (Fig. 3a–c, Supplementary Fig. 9). As individual microtubules have a defined diameter of 25 nm these structures serve as an experimental benchmark for SRM[24, 28, 46]. Simulations on nanobody labeling of microtubules using a maximal probe displacement of 5 nm and a localization precision cutoff of 10 nm have yielded an apparent fiber width of about 40 nm[24], which is in perfect agreement to our measured fiber width of 38.2 ± 9.2 nm (Supplementary Fig. 9a). Moreover, a detailed analysis of individual actin fibers comprising transiently expressed $_{BC2T}$actin showed comparable labeling densities as previously obtained for vimentin$_{BC2T}$ (Supplementary Fig. 9b-d).

Additionally, we used our approach to visualize non-structural proteins, namely the autophagosomal marker protein LC3B and the extracellular membrane marker GFP-GPI[47, 48]. To monitor induction of autophagy, we co-expressed $_{BC2T}$LC3B and GFP-LC3B in HeLa or A549 cells followed by incubation with DMSO or rapamycin to induce autophagosome formation. Wide-field imaging of chemically fixed cells, stained with bivBC2-Nb$_{AF647}$, showed a clear co-localization of GFP and nanobody signals at defined spots in rapamycin-treated cells, indicating correct localization of BC2-tagged LC3B at autophagosomes[47, 49]

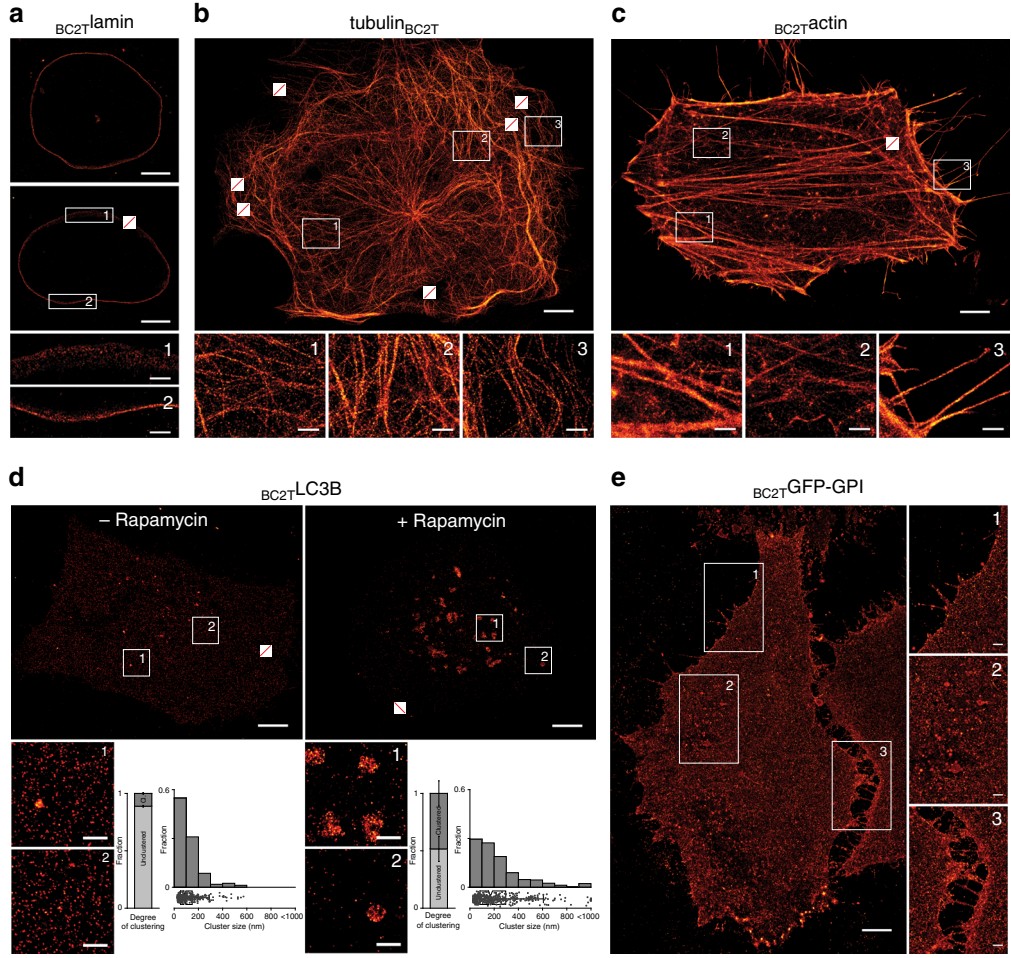

**Fig. 3** Super-resolution imaging of transiently expressed BC2-tagged proteins in chemically fixed cells. Representative dSTORM images of a **a** HeLa cell expressing $_{BC2T}$lamin, **b** U2OS cell expressing tubulin$_{BC2T}$ (filament width statistics in Supplementary Fig. 9a), **c** HeLa cell expressing $_{BC2T}$actin (coverage statistics in Supplementary Fig. 9b), **d** HeLa cells expressing $_{BC2T}$LC3B either left untreated or treated with rapamycin. Bar charts represent the degree of clustering, given as a relative fraction of cluster points versus noise points, errors given as standard deviation (S.D.). Histograms represent cluster diameters as determined by DBSCAN analysis with a bin size of 100 nm (x-axis) plotted against relative fraction (y-axis). Full data are represented underneath the histograms as box + scatter plots with the same x-axis. The box marks the three quartiles and the whiskers mark 95% of all the data. Total number of clusters $n = 342$ in non-treated cells, $n = 405$ in treated cells, $N = 3$ cells for untreated cells and $N = 4$ cells for rapamycin-treated cells (Supplementary Fig. 11), and **e** HeLa cells expressing $_{BC2T}$GFP-GPI. All cells were stained with bivBC2-Nb$_{AF647}$ (Methods section). Scale bars, images 5 μm, insets 1 μm. Crossed out rectangles mark the position of fiducial markers used for drift correction. Image reconstruction details are given in Methods section

(Supplementary Fig. 10a). Further, dSTORM imaging and subsequent DBSCAN cluster analysis of newly formed autophagosomes in $_{BC2T}$LC3B expressing cells after incubation with rapamycin revealed diameters of ~0.3–1.0 μm for these foci (Fig. 3d, Supplementary Fig. 11, Methods section), which is in accordance to previous findings[50, 51]. For BC2-tagged GFP-GPI ($_{BC2T}$GFP-GPI), we observed a clear co-localization of the nanobody and the GFP signal at the plasma membrane in chemically fixed HeLa cells (Supplementary Fig. 10b). Notably, with dSTORM, we detected a defined spatial organization of $_{BC2T}$GFP-GPI, e.g., the formation of small clusters and enrichment of $_{BC2T}$GFP-GPI molecules at cell–cell contacts, compared to a diffraction-limited, homogenous distribution observable by wide-field microscopy (Fig. 3e, Supplementary Fig. 10b).

**Visualization of endogenous proteins with the bivBC2-Nb.** To utilize the BC2-tag as an endogenous marker under native promotor expression, we first replaced the gene coding for the

nuclear DNA-binding protein cbp1 at its endogenous loci in the fission yeast *Schizosaccharomyces pombe* (*S. pombe*) by a C-terminally BC2-tagged version (cbp1$_{BC2T}$). Cells expressing cbp1$_{BC2T}$ show growth rates comparable to wild type (wt) and exhibit no morphological changes (Supplementary Fig. 12a, b). As *S. pombe* possesses a thick cell wall and a highly packed cellular environment, any immunofluorescence-based SRM approach suffers from high-unspecific background and low-staining quality. Notably, by staining endogenously expressed cbp1$_{BC2T}$ utilizing the bivBC2-Nb$_{AF647}$ we were now able to visualize an endogenous nuclear protein in *S. pombe* by dSTORM imaging (Supplementary Fig. 12a). Second, we stably introduced the coding sequence of the BC2-tag under the native β-actin promotor at the 5′-end of the first exon of endogenous β-actin in HeLa and A549 cells using the CRISPR/Cas9 technology. After monoclonal selection of cells exhibiting a heterozygous integration of $_{BC2T}$actin (HeLa-$_{BC2T}$ACTB; A549-$_{BC2T}$ACTB, Methods section) we treated both cell lines with transforming growth

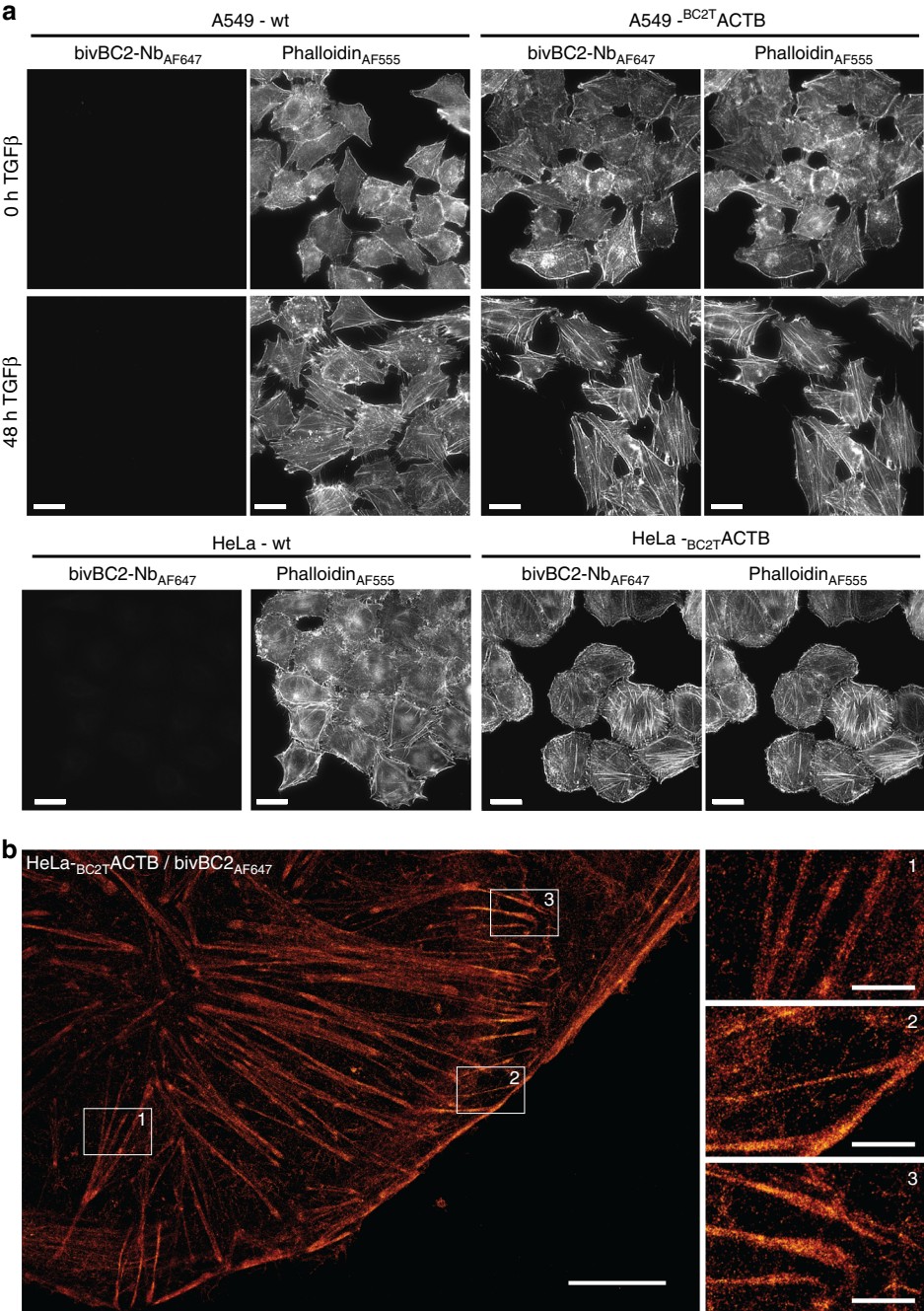

**Fig. 4** Visualization of endogenously expressed BC2-tagged actin labeled with bivBC2-Nb$_{AF647}$. **a** Wide-field images of chemically fixed wild-type A549 and HeLa (-wt; left panel), as well as chemically fixed A549-$_{BC2T}$ACTB and HeLa-$_{BC2T}$ACTB cells (right panel). Cells were either left untreated (0 h) or stimulated for 48 h with TGFβ (5 ng ml$^{-1}$) followed by staining with phalloidin$_{AF555}$ and bivBC2-Nb$_{AF647}$. Scale bars, 25 μm. **b** dSTORM image of a representative HeLa-$_{BC2T}$ACTB cell. Scale bars, image 5 μm, insets 1 μm. Image reconstruction details are given in Methods section. Imaging sequence taken from raw data acquisition can be found in Supplementary Movie 5, assessment of AF647 photophysics under dSTORM imaging conditions can be found in Supplementary Fig. 15

factor β (TGFβ) and monitored the induction of actin stress fibers by co-staining with bivBC2-Nb$_{AF647}$ and phalloidin$_{AF555}$. As expected, we detected the formation of stress fibers only in A549 wt and A549-$_{BC2T}$ACTB cells, which are described to respond to TGFβ[52] (Fig. 4a). dSTORM imaging further allowed a detailed insight into the non-disturbed actin network (Fig. 4b). These findings indicate that BC2-tagging of endogenous proteins is a viable approach for SRM studies to visualize cellular targets at endogenous levels and with minimal functional interference.

**bivBC2-Nb visualizes its target proteins in living cells**. To realize the advantages of the BC2-tag/bivBC2-Nb system also for live-cell applications, we first performed time-lapse imaging of HeLa cells transiently expressing $_{BC2T}$GFP-GPI. After addition of bivBC2-Nb$_{AF647}$ to the imaging medium, we observed a fast recruitment of the nanobody to its membrane-located antigen (Supplementary Fig. 13a), with a saturation of the nanobody signal within 20–30 min (Supplementary Fig. 13b). Single-particle tracking dSTORM imaging further allowed us to trace the highly

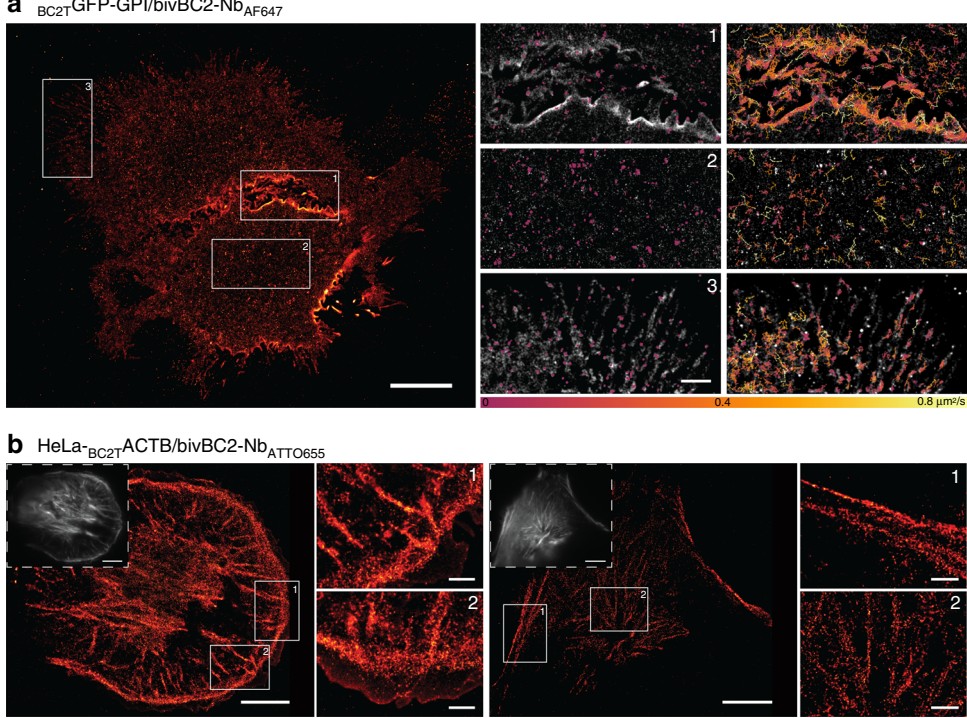

**Fig. 5** Super-resolution imaging and single-particle tracking in live HeLa cells. **a** dSTORM image of live $_{BC2T}$GFP-GPI expressing HeLa cells stained with bivBC2-Nb$_{AF647}$ and insets in gray scale overlaid with single-particle trajectories of immobile (diffusion coefficient below 0.02 μm$^2$ s$^{-1}$, left) and mobile (diffusion coefficient above 0.02 μm$^2$ s$^{-1}$, right). Color scale of diffusion coefficients is given under insets. Scale bars 10 μm in images, 2 μm in insets. Supplementary Movies 1–4 show the recorded live $_{BC2T}$GFP-GPI dynamics of the whole cell and the corresponding insets. **b** Live-cell dSTORM images of two HeLa-$_{BC2T}$ACTB cells stained with bivBC2-Nb$_{ATTO655}$. Wide-field fluorescence images in upper left corners. Scale bars in 10 μm, 2 μm in insets. Image reconstruction details are given in Methods section. Imaging sequence taken from raw data acquisition can be found in Supplementary Movie 6, assessment of ATTO655 photophysics under live cell dSTORM imaging conditions can be found in Supplementary Fig. 15

dynamic movements of thousands of $_{BC2T}$GFP-GPI molecules along the plasma membrane in high spatial and temporal resolution, e.g., the increased dynamics at cell-to-cell contact areas (Fig. 5a, Supplementary Movies 1–4). Second, for bivBC2-Nb staining of intracellular targets in living cells, we adapted a lipid-based protein transfection protocol[53, 54] and introduced bivBC2-Nb conjugated to different dyes into our cell lines HeLa-$_{BC2T}$ACTB and A549-$_{BC2T}$ACTB. Within 2 h, we observed cellular uptake of nanobody and its accumulation at the actin cytoskeleton irrespective of the attached fluorophore (Supplementary Fig. 14a). Prolonged time-lapse imaging of live HeLa-$_{BC2T}$ACTB cells for 5 h revealed a stable staining of the cytoskeleton (Supplementary Fig. 14b). Finally, we performed live-cell dSTORM imaging replacing AF647 with the fluorophore ATTO655 which is currently one of the few organic dyes exhibiting sufficient photoswitching under physiological intracellular conditions[8]. Two hours after nanobody transduction, dSTORM imaging of the bivBC2-Nb$_{ATTO655}$ staining revealed the intracellular actin network of HeLa-$_{BC2T}$ACTB cells (Fig. 5b) with sub-diffraction details as previously seen for chemically fixed HeLa-$_{BC2T}$ACTB cells (Fig. 4b). We further documented the photophysical differences in performance of the ATTO655 fluorophore, which is outcompeted by AF647 both in blinking statistics and brightness (Supplementary Fig. 15, Supplementary Movies 5 and 6). In summary, these data demonstrate that the bivBC2-Nb is functional within living cells where it retains its outstanding binding capacities allowing for live-cell dSTORM imaging. Other recently reported protein transduction approaches offer numerous alternatives to introduce the bivBC2-Nb in

various cell types for live-cell imaging of BC2-tagged proteins[41, 55–61].

## Discussion

In this study, we developed and extensively characterized a broadly applicable and transferable labeling strategy based on a structurally minimal tag in combination with the first peptide-specific nanobody suitable for SRM. In contrast to the widely established GFP/GFP-Nb system[28, 62] for dSTORM imaging of fusion proteins comprising a large fluorescent moiety, the short and inert BC2-tag allows an efficient and dense incorporation of ectopically and endogenously expressed proteins into higher-ordered cellular structures in mammalian and yeast cells. In particular, it does not interfere with the native organization of structural proteins, such as vimentin, lamin, actin, and tubulin known to be easily compromised by large protein tags. We demonstrated that all tested BC2-tagged proteins are efficiently detected with a high affinity, bivalent nanobody format, robustly labeled in a one-to-one nanobody to fluorophore ratio exhibiting a high-labeling efficiency competing with covalent detection systems. Together, this enables high-labeling coverage with minimal linkage error and thus allows for genuine SRM studies of physiologically undisturbed cellular structures. Moreover, the bivBC2-Nb is functional in living cells where it retains its binding capacity and labels its target structures over extended time periods, which renders the BC2-tag/bivBC2-Nb labeling system into a versatile tool for SRM imaging of fixed and living cells.

## Methods

**Expression constructs**. All primer sequences used for cloning are listed in a Supplementary Table 1. The expression construct coding for vimentin N-terminally fused to eGFP (GFP-vimentin) was previously described[41]. For generation of a photoactivatable (PA) mCherry (PAmCherry) fusion construct of vimentin (PAmCherry-vimentin) the coding sequence of PAmCherry was PCR-amplified from the pBAD/HisB-PAmCherry1 vector using the following primer set: PAmCherry-F and PAmCherry-R. The PCR product was purified, digested with the restriction enzymes AgeI and BglII and ligated in the AgeI/BglII sites of a vector coding for mCherry-vimentin thereby replacing mCherry with PAmCherry. An expression construct coding for vimentin with a C-terminal BC2-tag (vimentin$_{BC2T}$) was generated by the replacement of the mCherry sequence from a mCherry-Vimentin-BC2T (mCherry-vimentin$_{BC2T}$) fusion construct previously described[31]. Thus, vimentin$_{BC2T}$ cDNA was PCR-amplified using the primer VIM-BC2T-for and ligated into NheI and BamHI restriction sites of the template construct. Constructs coding for vimentin C-terminally fused to PAmCherry (vimentin-PAmCherry) and vimentin C-terminally fused to eGFP (vimentin-GFP) were generated by Gibson assembly of the three following fragments: fragment 1-pEGFP-N1 vector backbone digested with NheI and BsrGI, fragment 2- vimentin amplified from vimentin$_{BC2T}$ with the primer set VIM-for and VIM-rev, fragment 3- PAmCherry amplified from PAmCherry-vimentin or eGFP amplified from GFP-vimentin using the primer set PAmCherry/eGFP-for and PAmCherry/eGFP-rev. Fragments were assembled using the Gibson-Assembly Master Mix (New England Biolabs, cat. #E2611) according to the manufacturer's protocol. An expression construct coding for BC2-tagged β-actin ($_{BC2T}$actin) was generated by the combination of two PCR fragments derived from an eGFP-actin construct previously described in ref.[62]. The first PCR fragment was generated using primer BC2TActb(1)-for and BC2TActb(1)-rev. The second PCR fragment was generated using primer BC2TActb(2)-for and BC2TActb(2)-rev. Both DNA fragments were purified and ligated by compatible sticky ends generated by BssHII and SexAI restriction enzymes. To generate a BC2-tagged laminB1 ($_{BC2T}$lamin) expression construct the lamin B1 cDNA was PCR-amplified from a GFP-Lamin B1 DNA template[62] by PCR using primer BC2TLamin-for and BC2TLamin-rev and cloned into XhoI and NheI restriction sites of pEGFP-N1 vector thus introducing the BC2-tag sequence. The expression construct coding for C-terminally BC2-tagged lamin (lamin$_{BC2T}$) was generated by the replacement of the vimentin cDNA of the above described vimentin$_{BC2T}$ by lamin cDNA PCR-amplified from GFP-Lamin B1 using the primer laminBC2T-for and laminBC2T-rev. DNA fragments were purified and ligated by compatible sticky ends generated by NheI and BssHII restriction enzymes. The expression construct coding for C-terminally BC2-tagged tubulin (tubulin$_{BC2T}$) was generated by substituting vimentin cDNA of the above described vimentin$_{BC2T}$ with the tubulin cDNA PCR-amplified from pPAmCherry-tubulin (addgene 31930) with primers tubulinBC2T-for and tubulinBC2T-rev using restriction enzymes NheI and BssHII.

The expression construct coding for N-terminally BC2-tagged tubulin ($_{BC2T}$tubulin) was generated by substituting actin cDNA of the above described $_{BC2T}$actin with the tubulin cDNA PCR-amplified from pPAmCherry-tubulin (addgene 31930) with primers BC2Ttubulin-for and BC2Ttubulin-rev using restriction enzymes BssHII and BamHI. Mammalian expression construct coding for GFP-LC3B fusion protein, was generated by insertion of LC3B cDNA into BglII and EcoRI restriction sites of pEGFP-C1 expression vector. The expression construct coding for N-terminally BC2-tagged LC3B ($_{BC2T}$LC3B) was generated by substituting actin cDNA of the above described $_{BC2T}$actin with the LC3B cDNA PCR-amplified from GFP-LC3B using primers BC2TLC3B-for and BC2TLC3B-rev using restriction enzymes BssHII and BamHI. BC2T-tagged GFP-GPI construct for mammalian expression was generated by insertion of synthetic DNA fragment with an ORF coding for signal peptide of human CD59 (aa 1–25), BC2-Tag, eGFP and amino acids 92–128 of huCD59, which contains the GPI attachment site at aa 102 into BglII and NotI restriction sites of pEGFP-N2 vector DNA. Plasmids coding for Ypet-FtnA and SNAP-FtnA were purchased from addgene (cat. #98280 and 98282). To generate the expression construct coding for $_{BC2T}$Ypet-FtnA, BC2T was inserted to the N-terminus of Ypet-FtnA by PCR amplification of the complete Ypet-FtnA plasmid using BC2TYpet-for and BC2TYpet-rev. Subsequent recircularization of the amplified plasmid was performed using BssHII restriction sites.

All generated expression constructs were confirmed by sequencing and SDS-PAGE followed by western blot analysis using antibodies directed against eGFP (ChromoTek, cat. #3H9, dilution 1:1000), mCherry (ChromoTek, cat. #6G6, dilution 1:4000) or a BC2-Nb coupled to Alexa Fluor 647 (Thermo Fisher Scientific, cat. #A20006) (BC2-Nb$_{AF647}$, dilution 1:200) as previously described[31].

Bacterial expression vectors coding for the GFP-nanobody (GFP-Nb), the BC2-nanobody (BC2-Nb), and the bivalent BC2 nanobody (bivBC2-Nb) were provided by ChromoTek with a corresponding material transfer agreement. The bacterial expression construct of the bivalent VB6 nanobody (bivVB6-Nb) was previously described[41]. For all three constructs the original tag was replaced by a Sortase-tag (GSLPETG) upon PCR amplification of the full plasmid using the forward primer SorTag-Ins_for and SorTag-Ins_rev and subsequent recircularization using terminal AgeI restriction sites. The resulting expression constructs were confirmed by sequencing and bacterial expression followed by SDS-PAGE and immunoblot analysis using a C-terminal anti-His antibody (ThermoFisher Scientific, cat. #R930-

25, dilution 1:1000). An expression vector coding for SortaseΔ59 (pET28a-SrtAdelta59) was a gift from Hidde Ploegh (Addgene plasmid #51138)[63].

**Recombinant protein production and nanobody labeling**. GFP-Nb, bivVB6-Nb, BC2-Nb, and bivBC2-Nb all comprising a C-terminal Sortase-tag were expressed and purified as previously described[41, 64] and stored at −80 °C or immediately used for labeling. SortaseΔ59 was expressed and purified as described[63]. Alexa Fluor 647 (AF647)-coupled peptide H-Gly-Gly-Gly-Doa-Lys-NH$_2$ (sortase substrate) was purchased from Intavis AG. Chemical dye conjugation of BC2-Nb or bivBC2-Nb was carried out as described previously[31]. Briefly, purified nanobody was labeled with the N-hydroxysuccinimide (NHS) ester activated AF647 (ThermoFisher Scientific, cat. #A20006) according to manufacturer's guidelines. After coupling, unbound dye was removed by separation on Zeba Spin Desalting Columns (ThermoFisher Scientific, cat. #89890). For analysis, 0.1 μg of nanobodies were subjected to SDS-PAGE and analyzed on a Typhoon Trio (GE-Healthcare, excitation 633 nm, emission filter settings 670 nm BP 30) and subsequent Coomassie staining.

Degree of labeling (DOL, dye-to-protein ratio) was determined by absorption spectroscopy according to the instructions provided by ThermoFisher Scientific (stated for cat. #A20173). For BC2-Nb and bivBC2-Nb NHS-conjugated nanobodies DOLs of 1.8 ± 0.5 and 2.1 ± 0.7 were determined.

Sortase coupling of nanobodies was performed as previously described[33]. Briefly, 25 μM nanobody, 75 μM dye-labeled peptide dissolved in sortase buffer (50 mM Tris, pH 7.5, and 150 mM NaCl) and 100 μM sortase were mixed in coupling buffer (50 mM Tris, pH 7.5, 150 mM NaCl, and 10 mM CaCl$_2$) and incubated for 5 h at 25 °C. Uncoupled nanobody and sortase were depleted using Ni-NTA resin (Biorad, cat. #1560131). Unbound dye was removed using Zeba Spin Desalting Columns (ThermoFisher Scientific, cat. #89890). The dye-labeled protein fraction was analyzed by SDS-PAGE followed by fluorescent scanning on a Typhoon Trio (GE-Healthcare, excitation 633 nm, emission filter settings 670 nm BP 30) and subsequent Coomassie staining. For all sortase-coupled nanobodies DOLs of 0.7 ± 0.15 were determined.

**Bio-layer interferometry (BLI)**. The dissociation constants of BC2-Nb and bivBC2-Nb were determined on BLItz system (Pall ForteBio). Synthetic BC2 and BC2-BC2 (with (GGGGS)$_2$ linker) peptides with an N-terminal biotin-DoA-DoA-linker (Intavis AG) were immobilized on Streptavidin (SA) dip and read biosensors (Pall ForteBio, cat. #18-5020) using a concentration of 50 μM. For kinetic measurements of BC2- or bivBC2-Nbs three concentrations (120 nM, 240 nM, and 480 nM) of the Nbs in diluent buffer (1× PBS, 0.1% (w/V) BSA (Carl Roth, cat. #8076), 0.1 % (w/V) Triton X-100 (Sigma-Aldrich, cat. #T8787)) were used. Each measurement was done in duplicates with an association time of 180 s followed by 240 s dissociation in diluent buffer. Kinetic constants were determined using BLItz software (BLItz Pro 1.2, Pall ForteBio) according to global fitting of data sets.

**Cell culture and transfection**. The HeLa Kyoto cell line (Cellosaurus no. CVCL_1922) was obtained from S. Narumiya (Kyoto University, Japan), and the A549, U2OS and COS-7 cell lines were obtained from ATCC (CCL-185, HTB-96, CRL-1651). All cell lines were tested negative for mycoplasma using the PCR mycoplasma kit Venor GeM Classic (Minerva Biolabs, cat. #11-1025) and the Taq DNA Polymerase (Minerva Biolabs, cat. #53-0100). Since this study does not include cell line specific analysis, all cell lines were used without additional authentication. HeLa Kyoto, U2OS and COS-7 cells were cultured in DMEM +GlutaMAX (Life Technologies, cat. #31966-021) supplemented with 10% FCS (Life Technologies, cat. #10270-106) and 1 unit ml$^{-1}$ pen/strep (Life Technologies, cat. #15140-122). A549 cells were cultured in DMEM/F-12 (1:1) (Life Technologies, cat. #21331-020) supplemented with 10% FCS (Life Technologies, cat. 10270-106), 1 unit ml$^{-1}$ pen/strep (Life Technologies, cat. #15140-122) and 2 mM L-glutamine (Life Technologies, cat. #25030-024). Cells were trypsinized for passaging and cultivated at 37 °C in a humidified chamber with a 5% CO$_2$ atmosphere. Transient transfection of HeLa Kyoto, U2OS, and COS-7 cells with Lipofectamine 2000 (ThermoFisher Scientific, cat. #11668019) and transfection of A549 cells with Lipofectamine LTX (ThermoFisher Scientific, cat. #15338100) was carried out according to manufactures instruction.

**S. pombe strain construction**. The cloning strategy for BC2-tagging of the CBP1 gene at the C-terminus was adapted from[65]. The Saccharomyces cerevisiae ADH1 terminator and kanamycin resistance gene were amplified from the PAW8 plasmid[66] using the following primer pair F_KanR_BC2 and R_KanR. ~250 bp sequences up- and down-stream of the cbp1 gene were amplified from purified S. pombe DNA, with the primer pairs F1_cbp1, cbp1_BC2_R1, and F2_cbp1, R2_cbp1. Primers were designed to generate PCR products with overlapping regions of at least 20 bp. DNA fragments were assembled with overlap-extension PCR[67], using melting temperatures of the overlapping regions as the annealing temperature. All PCRs were performed with Q5 High-Fidelity DNA polymerase (New England Biolabs, cat. #M0491L). Volume of 10 μl of the PCR product was transformed into wild-type S.pombe using the Frozen-EZ Yeast Transformation II Kit (Zymo Research, cat. #T2001), plated onto YES agar plates and incubated overnight at 30 °C, then replica plated onto 200 μg ml$^{-1}$ G418 (Thermo Fisher

Scientific,) YES agar plates and incubated at 30 °C until single colonies were visible. Genomic integration was confirmed by colony PCR and DNA sequencing (Eurofins).

**S. pombe cell culture.** S.pombe was grown in YES medium (5 g yeast extract, 30 g glucose, 225 mg of each l-adenine, histidine, leucine, uracil, lysine hydrochloride in 1 l of Milli-Q water) at 30 °C overnight, then inoculated into fresh YES to a starting $OD_{600}$ of 0.1, grown to an OD600 of 0.4 and collected by centrifugation. Pellets were washed once with PEM buffer (100 mM Pipes, 1 mM EGTA, 1 mM $MgSO_4$, pH 6.9), then fixed for 15 min in 3.7% paraformaldehyde (PFA, Sigma-Aldrich, cat. #F8775) in PEM, washed $3 \times 10$ min with PEM containing 100 mM $NH_4Cl$ to quench the fixation, then permeabilized with a 1:1 mixture of methanol and acetone at $-20$ °C for 10 min. Fixed and permeabilized cells were then washed $3 \times$ with PEMBAL buffer (PEM + 1% BSA, 0.1% $NaN_3$, 100 mM lysine hydrochloride) and incubated in PEMBAL overnight. Before staining, cells were blocked with Image-iT FX signal enhancer (ThermoFisher Scientific, cat. #I36933) for 1 h, then stained for 48 h at 4 °C in PEMBAL containing ~0.5 µg ml$^{-1}$ of bivBC2–Nb$_{AF647}$ and 0.1% Triton X-100 (Sigma-Aldrich, cat. #T8787). Finally, cells were washed 2× with PEM containing 0.1% Tween-20 (Sigma-Aldrich, cat. #P7949), post-fixed with 4% PFA (Sigma-Aldrich, F8775) and 0.25% (w/V) glutaraldehyde (Sigma-Aldrich, cat. #G5882) in PEM for 10 min, then washed 2× with PEM and immobilized on poly-L-lysine coated Ibidi 8-well glass bottom slides (Ibidi GmbH, cat. #80826), previously cleaned with a 2% solution of Hellmanex III (Helma Analytics).

**CRISPR/Cas9D10A expression vector construct and HDR template.** Paired sgRNAs were designed for the ACTB (actin beta, Homo sapiens; PubMed Gene ID: 60) target gene locus using an online CRISPR gRNA design tool[68, 69] and synthesized as Ultramer DNA Oligonucleotides (Integrated DNA Technologies), ACTB_sgRNA and ACTB_HDR (Supplementary Table 1).

Next, paired sgRNAs were cloned according to a previously described procedure into a plasmid harboring Cas9D10A and a puromycin resistance cassette[70]. Briefly, the sgRNA fragment were PCR amplified using sgRNA_fw and sgRNA_rev primer thereby adding 3′ and 5′ domains homologous to plasmid encoded hU6 promotor and gRNA scaffold sequences, respectively. The 148 bp PCR amplicon was gel-purified and ligated at a 3:1 ratio to the 415 bp fragment generated by BbsI digest of the pDonor_U6 plasmid (a gift from Andrea Ventura, Addgene plasmid #69312)[70] using the NEBuilder Cloning Kit (New England Biolabs, cat. #E5520S). After treatment with Exonuclease RecBCD (New England Biolabs, cat. # M0345L) the column purified DNA plasmid was digested over night at 37 °C with BbsI. The linearized plasmid was then ligated into the BbsI-digested and dephosphorylated pSpCas9n(BB)-2A-Puro plasmid (a gift from Feng Zhang, Addgene plasmid #62987)[71] using T4 DNA ligase (NEB). Single-clone derived DNA plasmids were purified using QIAGEN Plasmid Midi Kit (Qiagen, cat. #12145) and verified by sequencing. The homology directed repair (HDR) template ACTB_HDR was synthesized as Ultramer DNA Oligonucleotides (IDT). HDR templates encoded for the HDR insert carrying the intended BC2-tag knock-in mutation flanked by left and right homology arms (each 50 bp) homologous to the ACTB target gene locus.

**Generation of BC2-tag knock-in cell lines.** $1 \times 10^6$ Hela Kyoto and A549 cells, respectively, were co-transfected at 50% confluency with 6.5 µg ACTB_HDR template oligonucleotide and 5.5 µg cloned Cas9N_Puro_ACTB_sgRNA expression vector construct or pEGFP plasmid (to control for transfection efficiency), respectively. For transfection of Hela cells Lipofectamine 2000 (ThermoFisher Scientific, cat. #11668019) and for A549 cells Lipofectamine LTX reagent (Thermo Fisher Scientific, cat. #15338100) was used. After 24 h, cells were trypsinized and re-plated in culture medium containing 1 µg ml$^{-1}$ puromycin dihydrochloride (Sigma-Aldrich, cat. #P8833). Forty-eight hours later mock-transfected control cells were completely killed by the antibiotic. Puromycin-resistant cell pools were expanded for 1 week and subsequently used for detection of CRISPR/Cas9D10A-induced BC2-tag knock-in by immunofluorescence staining using the bivBC2-Nb$_{AF647}$ and genomic PCR. Monoclonal knock-in cell lines were derived from cell pools showing successful BC2-tag knock-in by limiting dilution.

**Genomic PCR of BC2-tag knock-in cells.** Genomic DNA was isolated from puromycin-resistant cells using the QIAamp DNA Mini Kit (Qiagen, cat. #12125) according to manufacturer's instructions. Quantity 500 ng of purified genomic DNA was used for PCR amplification of the integrated BC2-tag sequence from the ACTB target gene locus using the primer ACTB_fw and BC2_rev. PCR products were separated on 1.5% agarose gels and visualized using ethidium bromide staining.

**Immunofluorescence staining for wide-field microscopy.** For immunofluorescence staining ~$1.5 \times 10^4$ HeLa Kyoto, U2OS cells, COS-7, or A549 cells per well of an 8-well µ-slide (Ibidi GmbH, cat. #80826) were plated. Next day, cells were transfected with plasmids coding for GFP-vimentin, mCherry-vimentin$_{BC2T}$, PAmCherry-vimentin, vimentin-GFP, vimentin-PAmCherry, vimentin$_{BC2T}$, $_{BC2T}$actin, $_{BC2T}$lamin, lamin$_{BC2T}$, $_{BC2T}$tubulin, tubulin$_{BC2T}$, GFP-LC3B, $_{BC2T}$LC3B, $_{BC2T}$GFP-GPI, and Ypet-FtnA or $_{BC2T}$Ypet-FtnA.

Twenty-four hours after transfection or in case of U2OS cells expressing $_{BC2T}$tubulin, tubulin$_{BC2T}$, Ypet-FtnA, or $_{BC2T}$Ypet-FtnA cells 72 h after transfection, cells were washed twice with PBS and fixed with 3.7% w/V paraformaldehyde (PFA) in PBS for 10 min at RT. Fixed cells were washed three times with PBS and permeabilized with a 1:1 mixture of methanol/acetone for 5 min at $-20$ °C. After three washing steps with PBS, cells were blocked with Image-iT FX signal enhancer (Thermo Fisher Scientific, cat. #I36933) for 30 min. Subsequently cells were washed with and stored in PBS until staining. For nanobody staining, GFP-Nb$_{AF647}$, BC2-Nb$_{AF647 (NHS)}$, BC2-Nb$_{AF647 (sort)}$, bivBC2-Nb$_{AF647 (NHS)}$, bivBC2-Nb$_{AF647 (sort)}$, or bivVB6-Nb$_{AF647}$ was added with a final concentration of ~50 ng ml$^{-1}$ in 5% BSA in TBS-T (0.05% (w/V) Tween) and incubated overnight at 4 °C. Unbound nanobodies were removed by three additional washing steps with TBS-T. Images were acquired with a MetaXpress Micro XL system (Molecular Devices) and ×40 magnification.

For staining of endogenous β-catenin, ~5000 HeLa Kyoto cells per well were seeded in a µclear 96-well plate (Greiner Bio One, cat. #655090). Cells were either transfected with an expression plasmid coding for vimentin$_{BC2T}$ or left untransfected. Twenty-four hours after transfection cells were further left untreated or continuously cultured in the presence of 10 µM CHIR99021 for 16 h. Subsequently, cells were washed twice with PBS and fixed with 3.7% w/V paraformaldehyde (PFA) in PBS for 10 min at RT. After three washing steps with PBS, cells were permeabilized and blocked with 0.1% Triton X-100 in 5% BSA in TBS-T for 30 min. For detection of endogenous β-catenin, cells were incubated with an anti-β-catenin antibody (BD Biosciences, cat. #610154, dilution 1:200) followed by detection with an Alexa Fluor 488 labeled anti-mouse-antibody (Invitrogen, cat. #A10680, dilution 1:1000).

**Generation of $_{BC2T}$Ypet-FtnA and SNAP-FtnA lysates.** Cell lysates comprising BC2-tagged FtnA oligomers were generated from transiently $_{BC2T}$Ypet-FtnA transfected U2OS cells and immobilized on Poly-L-lysine-coated 8-well µ-slides (Sigma-Aldrich cat. # P4707; Ibidi, cat. # 80826)[36].

**Staining of $_{BC2T}$Ypet-FtnA and SNAP-FtnA lysates.** Wells of an 8-well µ-slide (Ibidi, cat. #80826) containing lysates of $_{BC2T}$Ypet-FtnA and SNAP-FtnA expressing U2OS cells were blocked with 10% BSA in PBS for 30 min and with Image-iT FX signal enhancer (ThermoFisher Scientific, cat. #I36933) for an additional 60 min. Nanobodies were diluted to ~0.5 µg ml$^{-1}$ in staining (PBS, 10% BSA, 0.1% (V/V) Triton X-100 (Sigma-Aldrich, cat. #T8787)). SNAP-Surface Alexa Fluor 647 (New England Biolabs, cat. #S9136S) was diluted to 0.1 µM in the same solution. Volume of 200 µl of the staining solution was added to each well and stained for 6 h at room temperature. After staining, wells were washed three times for 15 min with PBS containing 0.1% Tween-20. The nanobody staining was post-fixed with 4% PFA and 0.25% glutaraldehyde in PBS for 5 min to make the binding permanent. In the case of SNAP, the binding is already covalent. Wells were then washed an additional three times for 15 min with PBS and imaged overlaid with 300 µl of PBS.

**Protein transduction.** HeLa_$_{BC2T}$ACTB, or A549_$_{BC2T}$ACTB were plated at ~5000 cells per well of a µclear 96-well plates (Greiner Bio One, cat. #655090) and cultivated at standard conditions. Next day, Nbs were transduced using Pro-DeliverIN (OZ Biosciences, cat. #PI10250) according to manufacturer's protocol. Per well of a 96-well plate 0.25 µl of Pro-DeliverIN was mixed with 0.75 µg Nb and incubated for 15 min at RT. Volume of 20 µl Opti-MEM (ThermoFisher Scientific, cat. #31985062) was added to the mixture and immediately transferred to the cell culture medium in the well. After 2 h, medium was replaced by imaging medium DMEM$^{gfp}$-2 (Evrogen, cat. #MC102) supplemented with 10% FCS, 2 mM L-glutamine and cells were imaged.

**Live-cell staining and imaging.** HeLa Kyoto transiently expressing $_{BC2T}$GFP-GPI, HeLa-$_{BC2T}$ACTB, or A549-$_{BC2T}$ACTB cells were plated at ~5000 cells per well of a µclear 96-well plate (Greiner Bio One, cat. #655090) and cultivated at standard conditions. Next day, time-lapse imaging was performed in a humidified chamber (37 °C, 5% CO2) of a MetaXpress Micro XL system (Molecular Devices) at ×40 magnification. For live-cell staining of $_{BC2T}$GFP-GPI, culture medium was replaced without washing by live-cell visualization medium DMEM$^{gfp}$-2 (Evrogen, cat. #MC102) supplemented with 10% FCS, 2 mM L-glutamine and 1 µg ml$^{-1}$ bivBC2-Nb$_{AF647}$. Time-lapse imaging with 4–5 min intervals was started immediately upon medium replacement. For live-cell staining of HeLa-$_{BC2T}$ACTB and A549-$_{BC2T}$ACTB upon protein transduction of nanobodies, cells were washed once with and placed in DMEM$^{gfp}$-2 medium 2 h after addition of transduction mix (see "protein transduction" section above) and imaged in hourly intervals.

**Quantification of staining intensities.** HeLa Kyoto cells were plated at ~5000 cells per well of a µclear 96-well plates (Greiner Bio One, cat. #655090) and transfected with expression plasmid for mCherry-VIM$_{BC2T}$. Next day, cells were fixed and stained with the same concentration (1 µg ml$^{-1}$) of monovalent or bivalent BC2-Nbs conjugated to AF647 either by NHS conjugation or via sortase coupling. To assess staining quality of the different nanobody formats we calculated the ratio of the staining intensity in mCherry-VIM$_{BC2T}$ expressing cells and in non-transfected cells (background). Staining intensities were determined using a

custom-written cell identification algorithm (MetaXpress, Custom module editor). In brief, transfected cells were identified based on cell size parameters and a threshold setting for mCherry fluorescence intensity above local background. Background fluorescence was defined as the average fluorescence of the remaining image area precluding mCherry-VIM$_{BC2T}$ expressing cells. For statistical significance the average fluorescence intensity of a large number of transfected cells was determined (BC2-Nb$_{AF647\ (NHS)}$: $n = 115$; bivBC2-Nb$_{AF647\ (NHS)}$: $n = 134$; BC2-Nb$_{AF647\ (sort)}$: $n = 150$; bivBC2-Nb$_{AF647\ (sort)}$: $n = 195$).

**Immunofluorescence staining for dSTORM imaging.** To achieve the higher labeling density required for dSTORM imaging the staining protocol was slightly modified. Cells were prepared the same way as described up to the storage step in PBS. After storage, cells were blocked with 10% (w/V) BSA (Carl Roth, cat. #8076) in PBS for 30 min, then additionally with Image-iT FX signal enhancer (ThermoFisher Scientific, cat. #I36933) for 60 min. Antibodies and nanobodies were diluted to ~0.5 µg ml$^{-1}$ in staining/permeabilization solution (PBS, 10% BSA, 0.1% (V/V) Triton X-100 (Sigma-Aldrich, cat. #T8787)). Conventional immunostaining was done at 4 °C for 24 h with the primary antibody (V9, mouse monoclonal, Sigma-Aldrich, cat. #347M-1)[72] followed by two washes with PBS, and then stained at 4 °C for 24 h with the secondary antibody (donkey-anti-mouse AF647, ThermoFisher Scientific, cat. #A-31571). For staining with nanobodies, cells were incubated at 4 °C for 48 h. Unbound Nbs were removed by two washes with PBS-T (0.1% w/V Tween-20 (Sigma-Aldrich, cat. #P7949)) and samples were post-fixed with 4% PFA (Sigma-Aldrich, cat. #F8775) and 0.25% (w/V) glutaraldehyde (Sigma-Aldrich, cat. #G5882) in PBS for 5 min to make the binding permanent. Finally, cells were washed twice with PBS to remove fixation solution and stored in PBS with 0.1% (w/V) sodium azide (Carl Roth, cat. #4221) until imaging.

**dSTORM imaging and post-processing.** A 1:5000 dilution of fluorescent beads (FluoSpheres 715/755, ThermoFisher Scientific, cat. #F8799) was sonicated to break up clumps of beads. Volume of ~5 µl of the beads were added to the sample and allowed to settle and adhere for 15 min, to serve as fiducial markers for drift correction. Images were recorded on a customized Nikon Ti-Eclipse inverted microscope, equipped with a CFI Apochromat TIRF ×100 objective with a numerical aperture of 1.49 (Nikon) and an iXON ULTRA 888 EMCCD camera (Andor). AF647 was imaged in 100 mM MEA (Sigma-Aldrich, cat. #M6500-25G) with a glucose oxidase (Sigma-Aldrich, cat. #G2133, C100) oxygen scavenger system[73]. The sample was illuminated with an OBIS LX 637 nm laser (Coherent) which was filtered through a ZET 640/10 bandpass, modulated by an Acousto-Optic Tunable Filter (Gooch & Housego, TF525-250-6-3-GH18) and focused by a ZET405/488/561/640 m dichroic mirror (Chroma) onto the back focal plane of the objective resulting in a final intensity of 2–4 kW cm$^{-2}$ in the sample. The readout was collected by blocking the laser light by the bandpass ZET405/488/561/640 and passing through a 689/23 nm single-band bandpass filter (All filters AHF Analysentechnik AG). For each dSTORM image reconstruction, 10,000–20,000 imaging frames with an exposure time of 70 ms were recorded at a pixel size of 129 nm. The camera, microscope and AOTF were controlled by µManager software[74] on a PC workstation. Single-molecule localizations were extracted from the movies with the open-source software Rapidstorm 3.2[75]. Drift correction was performed by custom written Python 2.7 algorithms that extract and correct for fluorescent bead tracks. NeNA as described in ref. [43] was done with the open-source software Lama[76] on a section of the image that contained no fiducial markers. Localizations appearing within the radius of the NeNA value on several frames were grouped into one localization using the Kalman tracking filter in Rapidstorm 3.2. Final images were reconstructed at a pixel size of 10 nm. For visualization, a Gaussian blur filter was applied in the ImageJ software using NeNA as the sigma value. dSTORM imaging in yeast cells was performed in PEM buffer containing 10 mM MEA (Sigma-Aldrich, M6500) and 1 mM methyl viologen dichloride hydrate (MV) (Sigma-Aldrich, cat. #856177).

**PALM imaging and post-processing.** For imaging of PAmCherry-vimentin, the sample was illuminated with an OBIS LX 561 laser (Coherent), filtered through a ZET 561/10 clean-up filter (AHD Analysentechnik, Germany) at an intensity of 800 W cm$^{-2}$ and an OBIS LX 405 laser (Coherent) at intensites of ~25–5000 mW cm$^{-2}$. The readout was collected through a 610/75 bandpass filter (AHF Analysentechnik, Germany). Twenty thousand imaging frames were recorded at an exposure time of 70 ms. The number of PAmCherry molecules activated each frame was kept at a steady rate by increasing 405 laser intensity until all of the PAmCherry was readout. All other microscope parameters and image post-processing remained the same as for dSTORM imaging.

**Super-resolution analysis of endogenous β–catenin staining.** To evaluate the effect of endogenous β-catenin staining on dSTORM imaging of low-abundance non-structural proteins, non-transfected HeLa and $_{BC2T}$LC3B expressing HeLa cells were fixed and stained with bivBC2-Nb$_{AF647}$ or GFP-Nb$_{AF647}$ for 48 h, imaged and post-processed as described in the "Immunofluorescence staining for dSTORM imaging" and "dSTORM imaging and post-processing" sections".

Analysis was performed on non-transfected HeLa cells stained with bivBC2-Nb$_{AF647}$ or GFP-Nb$_{AF647}$, in comparison to $_{BC2T}$LC3B-expressing cells stained with

bivBC2-Nb$_{AF647}$. The GFP-Nb$_{AF647}$ staining served as a baseline of non-specific nanobody binding. Three $15 \times 15$ µm ROIs for each condition were analyzed with DBSCAN (DBSCAN parameters: $\varepsilon = 40$ nm; MinPts $= 6$)[35] to identify single nanobodies. The density of nanobodies per µm$^2$ was calculated for each ROI and the means of each condition plotted as bar charts with S.D. as the error.

To evaluate the effect of endogenous β–catenin staining when imaging abundant structural proteins, non-treated and CHIR99021-treated non-transfected and vimentin$_{BC2T}$ expressing HeLa cells were stained with bivBC2-Nb$_{AF647}$ and imaged as described in the "Immunofluorescence staining for dSTORM imaging" and "dSTORM imaging and post-processing" sections. Localization counts for three cells per condition were obtained with the RapidSTORM software[75] and cell areas were measured in Fiji[77]. Localizations per µm$^2$ were calculated and the means of each condition plotted as bar charts with S.D. as the error.

**Imaging of $_{BC2T}$Ypet-FtnA and SNAP-FtnA lysates.** A 488 nm Sapphire laser (Coherent Inc., Santa Clara, California USA) was used to excite the Ypet and the readout was collected through a 525/50 single bandpass filter (AHF Analysentechnik AG). Otherwise, the microscope setup was the same as described in the dSTORM paragraph above. The red laser intensity was reduced to ~0.4 kW cm$^{-2}$, to avoid pixel saturation when imaging stained FtnA oligomers while still allowing for the detection of single-AF647 molecule blinking events. In the case of $_{BC2T}$Ypet-FtnA, fluorescent spots were identified by their Ypet signal. Since the SNAP-tagged FtnA oligomers lacked the Ypet signal, they were excited with very low 640 nm laser (>0.01 W cm$^{-2}$) and chosen within 2 s to avoid photobleaching. Both were then imaged for 200 frames with an exposure time of 50 ms.

**$_{BC2T}$Ypet-FtnA and SNAP-FtnA analysis.** The fluorescence microscopy movies were analyzed with a custom ImageJ script in the open access software Fiji[77]. AF647 single-molecule blinking events were identified in the last 100 frames of the movies. Molecules that blinked at least twice in the last 100 frames were selected in different quadrants of the imaged area. A round ROI with a diameter of 10 pixels was drawn around each molecule and the intensity trace of the ROI for all 200 frames was extracted. The integrated intensity value was plotted over time using the software OriginPro (Origin Lab Corp.). Only blinking events with a clear jump in the integrated intensity values co-inciding with a visual blink in the movie were measured. Intensity of blinking events was measured by calculating the difference between baseline and peak intensity of the event (Supplementary Fig. 3a). A total of $N = 157$ blinking events were measured. The single-molecule integrated intensity values were binned with a bin size of 100 AD counts, plotted as a relative frequency histogram and fitted with the log-normal distribution function in OriginPro (Origin Lab Corp.):

$$f(x) = \frac{1}{x\sigma\sqrt{2\pi}}\mathrm{e}^{-\frac{(\ln x - \mu)^2}{2\sigma^2}}$$

as described in ref. [78] yielding $\mu = 5.68$ and $\sigma = 0.4$ (Supplementary Fig. 3a). FtnA-oligomer spots were analyzed in a similar manner, with the integrated intensity value of the diameter of 10 pixels ROI of the second frame used for further analysis. A total of 130 $_{BC2T}$Ypet-FtnA spots and 89 SNAP-FtnA spots were analyzed. The integrated intensity data was binned with a bin size of 1000 AD counts and plotted as relative frequency histograms (Supplementary Fig. 3b). Approximating the labeling of FtnA oligomers with AF647 molecules with a binomial distribution, with the degree of labeling (0.7 ± 0.15 in case of the bivBC2-Nb$_{AF647}$ nanobody and 0.95 ± 0.05 in case of the AF647-BG) representing the p and the number of FtnA molecules in an oligomer representing the n parameter, we could calculate the probability for the occurrence of each labeling state (e.g., 24, 23, 22… AF647 per oligomer). Data sets assuming full labeling were simulated by calculating the linear combination of calibration distributions following the binomial mixture of visual oligomeric states due to the degree of labeling by recursively convolving the single-fluorophore log-normal distribution ($\mu = 5.68$, $\sigma = 0.4$) using a MatLab (Math-Works, Natick, MA-US)-based tool, as published and described in ref. [78]. To evaluate the completeness of labeling with the bivBC2-Nb$_{AF647}$ nanobody and the SNAP-tag, we then compared the medians of the simulated fully labeled scenario data sets with our measured distributions.

**Image analysis of dSTORM images.** A custom ImageJ script was used to measure the widths and coverages of vimentin, actin, and tubulin fibers on reconstructed non-blurred images. The segmented line tool in ImageJ was used to manually draw lines along vimentin filaments and line thickness was adjusted to fit the filament. Each image was divided into $10 \times 10$ µm sections and 15 random filaments were measured in each section. Line selections were straightened using the straighten tool and an intensity profile was plotted for each filament. To determine filament width, the intensity profile was fitted with a Gaussian curve and the resulting sigma value was multiplied with 2.35 to obtain the full width at half maximum (FWHM). To determine lengthwise coverage the image was converted into a binary image and the fraction of covered area of the middle 3 pixels was calculated. To reduce measurement error, line selections were wobbled by 0.5 pixels in four directions and values from all 5 measurements were averaged. To estimate image resolution, custom written Python 2.7 (Python Software Foundation) and Fiji[77] algorithms utilizing code from the Lama software[76] and GDSC SMLM ImageJ plugin (http://

www.sussex.ac.uk/gdsc/intranet/microscopy/imagej/gdsc_plugins) were used. NeNA localization precision[43] and Fourier Image Resolution (FIRE)[44] were calculated for all analyzed regions. To calculate FIRE, localization files were split into two by alternating frames ("random split" option unchecked). The threshold was set to Fixed 1/7 and the Fourier image scale was set to a constant 16 for all analyzed regions. Other parameters were left as default (auto image scale = 2048, sampling factor = 1). The results were plotted using the software OriginPro (Origin Lab Corp.).

**LC3B clustering analysis**. Imaging and post-processing of $_{BC2T}$LC3B cells was done as described in the "dSTORM imaging and post-processing" section. The obtained localization files were loaded into the MatLab-based software PALMsiever[79]. Density-based clustering analysis was performed with the density-based spatial clustering of applications with noise (DBSCAN) algorithm (DBSCAN parameters: $\varepsilon = 50$ nm; MinPts = 40)[35]. Three cells were analyzed for the untreated conditions and four cells for the rapamycin-treated conditions. To compare the degree of clustering, the ratio of non-clustered to clustered localizations was calculated for each cell. Points assigned as core and border points by DBSCAN were considered as clustered, while noise points were considered as non-clustered. Cluster size was calculated with a custom-written Python 2.7 script, measuring the average distance of cluster points from the cluster center of mass. Degree of clustering results were plotted as stacked bar charts, using S.D. as the error and the cluster sizes were plotted as histograms with a bin size of 100 nm, as well as individual data points and box + whisker plots, the ends of the box marking the 1st and 3rd quartiles, notch marking the median and whiskers encompassing 95% of the data in OriginPro (Origin Lab Corp.).

**Live cell dSTORM imaging**. HeLa Kyoto transiently expressing $_{BC2T}$GFP-GPI and HeLa-$_{BC2T}$ACTB were seeded in 8-well Ibidi μ-slides to an approximate density of 7500–15,000 cells per well, cultivated for 24 h at standard conditions then sealed in falcon tubes containing equilibrated culture medium. Upon arrival, cells were placed in an incubator for 3 h to recover and re-equilibrate. Meanwhile, 3 μl of Pro-DeliverIN was mixed with 9 μg of nanobody and incubated at RT for 15 min. After incubation, 60 μl of Opti-MEM were added to the mix and transferred into culture wells containing 300 μl of culturing medium. Cells were incubated for 2 h in the incubator, then washed two times with pre-equilibrated imaging medium DMEMgfp-2 without FCS. Volume of 300 μl of pre-equilibrated imaging medium without FCS, supplemented with the appropriate STORM buffer components were added to the wells. Well slide lids were sealed with parafilm and imaged on a custom built piezo-electric heating stage at 37 °C.

For imaging of $_{BC2T}$GFP-GPI, cells were stained with the bivBC2-Nb$_{AF647}$ and the imaging medium without FCS was supplemented with filter sterilized Tris-Hydrochloride to a final concentration of 100 mM and 10 mM MEA. The microscope setup remained as before. To record single-molecule tracks, 640 nm laser intensity was reduced to 1–2 W cm$^{-2}$ and 20,000 imaging frames were recorded with an exposure time of 40 ms. For imaging of $_{BC2T}$ACTB, cells were stained with the bivBC2-Nb$_{ATTO655}$ and the imaging medium without FCS was supplemented with 50 μM ascorbic acid. ATTO655 was excited with 2-4 W cm$^{-2}$ of 640 nm laser and 20,000 imaging frames were recorded with an exposure time of 50 ms. $_{BC2T}$ACTB images were processed as described in the "dSTORM imaging and post-processing" section.

**Live cell image processing and analysis**. Single-$_{BC2T}$GFP-GPI particles were tracked with the help of customized tracking software written in C++ and visualized by customized software written in C++, to filter for trajectories of at least 5 steps and group single-molecule localizations or trajectories by their apparent diffusion coefficient (as calculated by MSD analysis). Only trajectories with 5 or more steps were used for analysis and visualization. Single-particle trajectories were grouped into immobile (diffusion coefficient <0.02 μm$^2$ s$^{-1}$) and mobile particles (diffusion coefficient >0.02 μm$^2$ s$^{-1}$) and overlaid on top of a super resolution image reconstructed in Rapidstorm 3.2[75].

**General statistical analysis of vimentin, actin, and tubulin**. All labeling methods were repeated at least twice and imaged on at least two different days using the same setup and imaging parameters. Five cells from each labeling method were chosen based on the quality of drift correction. Cells were divided into 10 × 10 μm quadrants and 15 filaments in each quadrant were randomly chosen for analysis by at least two different people. All measured values were plotted as individual data points and box + whisker plots, the ends of the box indicating the first and third quartiles and the notch indicating the median. The whiskers encompass 95% of all the data. To confirm that the results of the different labeling methods differ significantly, vimentin filament widths were divided into a thin (0–75 nm), medium (75–150 nm), and thick (150 nm and above) category on which we performed a $\chi^2$-test of homogeneity. The total number of measured filaments was high enough to ensure adequate test power ($n$-values for all conditions given in Supplementary Fig. 4). $\chi^2$-values confirmed that the filament phenotypes for the labeling methods do differ at a 0.001 level of significance. To confirm that the ratio of fiber thicknesses does not significantly differ between bivVB6-Nb$_{AF647}$ and bivBC2-Nb$_{AF647}$, a $\chi^2$ homogeneity test was performed on the categories 0–150 nm width and 150 nm

and above, yielding a 0.001 level of significance. To test for correlation between FIRE, NeNA and coverage values, linear regression was performed pairwise on FIRE/NeNA, FIRE/coverage, and NeNA/coverage for all labeling methods. For FIRE/NeNA pairs, the slope of the linear correlation fits for bivVB6-Nb$_{AF647}$, GFP-Nb$_{AF647}$, and bivBC2-Nb$_{AF647}$ was significantly different from 0 at a 0.05 level of significance, while in the case of PAmCherry it was not. Correlating FIRE and NeNA to coverage, the slope of the linear fit was never significantly different from 0 at a 0.05 level of significance.

**Data availability**. The data that support the findings of this study are available from the corresponding authors upon reasonable request.

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

## Acknowledgements

The authors thank Christian Linke-Winnebeck and Benjamin Ruf (both ChromoTek GmbH) for technical support in nanobody production and labeling, Yana Parfyonova for cloning of the BC2T GFP-GPI construct and performing corresponding live cell studies, Maruša Kustec for help with general statistical tests, Urban Završnik for help with Python programming language scripting, Alex Herbert for help with his GDSC SMLM FIRE ImageJ Plugin and Knut Drescher for kindly providing his mammalian cell culture equipment. D.V., I.V., B.P.-L., and U.E. acknowledge funding by the Max Planck Society, SYNMIKRO and the Fonds der Chemischen Industrie. U.R., B.T., and J.M. gratefully acknowledge the Ministry of Science, Research and Arts of Baden-Württemberg (V.1.4.-H3-1403-74) for financial support.

## Author contributions

U.E. and U.R. conceived the study. D.V., B.T., J.M., P.D.K., M.B., C.S., I.V., B.P.-L., U.E., and U.R. performed all experiments, U.E. and U.R. wrote the manuscript with the input from all authors.

## Additional information

**Competing interests:** U.R. is shareholder of the company ChromoTek GmbH. All other authors declare no competing financial interests.

