## [Peer Review File · Nature Communications]

Reviewers' comments:

Reviewer #1 (Remarks to the Author):

The authors have followed up on their work that first reported the BC2 tag scheme and have shown the application of the tag for super-resolution imaging in fixed and live cells. The paper claims that the peptide specific nanobody tagging can be applied for super-resolution imaging. Data in the paper back up the claims reasonably well. While the work demonstrates finer details of the BC2 tag labeling and careful controls for SMLM, the paper is not suitable for publication in Nature Communications. The paper shows incremental improvement over the previous report (from the same authors) and does not provide any deeper understanding of the labeling, mechanistically or otherwise. Some relevant details are missing and it is not clear what the unique selling point of this approach is. The use of endogenous tagging supports the utility of this approach; however, it does not aid in underscoring the importance of the tagging scheme itself. Finally, there are several speculative/ unexplained results, particularly in the control experiments that have been reported using fluorescent proteins and tags. I recommend that the authors consider the points below for improving the manuscript and submit it to a more focused chemistry / chemical biology journal. Put differently, Nature communication publications typically deliver on a novel technology combined with a sound understanding of the mechanisms behind these observations. In many cases, such papers revolutionize the thinking/ experiments in a given field. In this case, both these aspects are missing and hence, I recommend the authors to consider alternative journals.

Here are some comments that can improve the quality of the manuscript:

Line #29-30 – The major limiting factor for the quality of LM techniques is the photons emitted from a fluorophore. This needs to be mentioned here with citation (Thompson et al., 2002)

Line 31-32 – There is a relatively new class of tags targeting fluorogens (See works of Lavis, Gautier, Bruchez, Moerner). These tags should be highlighted since some of them have been shown to work with SMLM based techniques.

65-67 – what do the authors attribute the inferior signal from the monovalent tagged Nb to?

80-85 – While the authors report signal to noise here, the critical parameter to be reported when comparing labeling strategies is photon numbers/ localization and/ or the localization precision. Additionally, the SNR is a function of the background as well. The authors must report the background photons/ pixel in both the cases.

Fig 2 – For each cell image shown, it must be mentioned if they are live/ fixed cells. Same is true about the image display. The authors must mention where ever there is a reconstruction (and outline the reconstruction type: scatter, histogram, Gaussian etc.)

104-106 – Is the phenotype here just smaller cells? Is this a live/ fixed cell comparison? What reasons do the authors attribute for this difference between GFP vs pa-mCherry labeling? Is there literature supporting this difference between two FPs? What if this is an artifact of fixation (see a recent paper from the Tijan lab where they discuss this)

124-133 – Chromophore maturation times must be on the same order for pa-mCherry and GFP. The authors must definitely perform a photoactivation light dose dependent experiment and infer the difference between GFP and pa-mCherry. This section is full of "likely" causes for experimental observations. The authors must provide some experimental evidence for some observations in the least.

One big gap in the understanding about the delivery of these tags that the authors speculate about

but do not clearly explain the delivery of Alexa647 labeled protein in the cells. It is known that Alexa647 is a membrane impermeable dye. It is possible that the Nb shields the dye charge by encapsulating/ forming some kind of secondary structure around the dye. One experiment that could be performed here is tagging dyes with different charges/ backbones (Cyanines vs Rhodamines, for example). The uptake for dyes with different charges/ backbones will help us further our understanding of the mechanism behind the uptake. This will also make the technology more general and widely used.

Minor but important:

Please read the SI carefully before submission. There are several small things that have been overlooked. This is another aspect that should be improved drastically for submission to any journal. Noting only 2 examples here:

SI Fig 1 – The authors have confused between k_{off} and K_d in the table and legend for figure 1 in SI. Please write k_{off} in the table.

SI line 104 – Peptide Dissolved not solved

It will be good to provide a table comparing properties such as labeling concentrations, incubation times, # of washes for the Nb tag and conventional tags.

Reviewer #2 (Remarks to the Author):

Here the authors describe a promising new bivalent labeling strategy compatible with single molecule localization microscopy (SMLM) using a structurally minimal tag. The potential impact of the method is significant, the labeling density and specificity in live cells is impressive even compared to many fixed samples, and the unmodified nanobody uptake is unexpected and certainly adds utility to this approach. My suggestions are primarily aimed at improving transparency regarding the flexibility and broad applicability of this technique, with the goal of enhancing impact and adaptation by the field.

Major comments –

1) The primary impact/novelty of the manuscript is an improvement in labeling affinity and completeness with a structurally minimal tag. In this context, the heavy reliance on comparatively abundant structural proteins, the traditional low-hanging fruit of super-resolution microscopy, is not appropriate. This is puzzling given that the authors present data on non-structural proteins, but relegate it to supplementary figures with minimal analysis. A comparative attempt to colocalize two proteins known to co-complex in low copy number would be a useful addition to the paper.

2) While the apparent uptake of nanobody into live cells for labeling purposes is extraordinary, treatment of this phenomenon should not be limited to images of immortalized cancer cell lines. Minimally there should be a quantitative treatment of the kinetics of this effect and this should be replicated in pombe; ideally I would like to see this on primary cells or tissue.

3) It appears that every BC2 construct used was constructed as a C-terminal fusion. This should be tested in other configurations; if C-terminal structure is required for nanobody recognition, this should be discussed in detail. It is not correct as author suggests in the summary that the native alpha-tubulin is unaffected by C-terminal additions. C-terminal post-translational modifications abound on tubulin, including multiple carboxy terminal residue cleavages. I find the assertion dubious for the other listed proteins as well in the absence of appropriate references. While a short peptide would be expected to have a less disruptive impact than a full fluorescent protein, the option to change sites can be critical to experiments and the feasibility of such a substitution

should be examined in detail.

Minor comments -

- 1) N values for testing categories of filament width are not apparent. Statistical sections would benefit from editing for clarity.
- 2) The methods section is otherwise exemplary, and demonstrates a commitment to reproducibility and broad adoption of new method. This level of detail is not common and I thank the authors for their diligence.
- 3) The antigenic peptide tag is not particularly novel, but important to overall paper; in this respect it needs better characterization and discussion.

Reviewer #3 (Remarks to the Author):

In this manuscript Virant et al. are reporting a novel labeling method for dSTORM based on nanobody that recognize a peptide tag. The authors are evaluating the labeling density and the impact of the probe size by comparing with commonly used labeling methods. Furthermore, applications visualizing endogenous proteins and targeting in living cells were also proposed. The reviewer recognized a potency of the novel labeling technique as high-density labeling with minimum size effect, and therefore the manuscript holds interests for general readers of Nature Communications. However, there are two concerns to be addressed prior to be published.

1) The new method is based on a nanobody, BC2-Nb, which binds to a peptide tag. BC2-Nb was originally developed as a nanobody against β -catenin that is endogenously expressing in various cells. Hence, the effect of endogenous β -catenin should be discussed in the manuscript. Expression levels of proteins shown by dSTORM in this manuscript, i.e. actin, vimentin, and tubulin, are known to be high and therefore endogenous β -catenin might be negligible in these cases, but the expression level of most of the protein of interest is much lower and this reviewer was wondering if the effect of endogenous β -catenin becomes obvious.

2) Labeling in living cells by BC2-Nb is interesting, and this reviewer was convinced that it worked for wide-field fluorescence imaging. However, apparently the background is rather high and most probably the labeling efficiency is not as good as the labeling in fixed cells. Therefore, this reviewer suspects if the labeling in living cells is not applicable for dSTORM imaging. If it is applicable, please simply show an example. If not, it should be clearly stated in the text and explain, otherwise audience might get confused that the method is applicable for dSTORM under living conditions.

Reviewer 1

The paper shows incremental improvement over the previous report (from the same authors) and does not provide any deeper understanding of the labeling, mechanistically or otherwise.

In the first submission we had demonstrated that improved labeling with the bivalent BC2-Nb is likely due to a reduced off-rate of the nanobody (**Supplementary Fig. 1**). To further elucidate this mechanistically, we now include a set of experiments for which we have generated constructs comprising either singular or tandem BC2-tags and performed biochemical binding studies using biolayer interferometry. These experiments show that the reduced off-rate is mediated by the binding of the bivalent BC2-Nb to a single BC2-tag and is not due to simultaneous binding of two BC2-epitopes. We added these results as a novel part to the **Supplementary Fig. 1** now shown as **new Supplementary Fig. 1d**.

We further assessed the completeness of structural labeling of the BC2-tag/bivBC2-Nb system by using the oligomeric bacterial protein FtnA as a protein standard with the defined number of 24 subunits. Our data show that the BC2-tag/bivBC2-Nb detection system competes with the efficiency of the covalent SNAP-tag system and is superior to efficiencies achievable with photoactivatable FPs. We added these results and corresponding references in the **main text**, **Fig 1c** and more details in new **Supplementary Fig. 3** and **Methods**.

Line #29-30 – The major limiting factor for the quality of LM techniques is the photons emitted from a fluorophore. This needs to be mentioned here with citation (Thompson et al., 2002)

We thank the reviewer for pointing us at this lapse and included the limit of photon counts per fluorophore with an appropriate statement and corresponding reference in the revised manuscript.

Line 31-32 – There is a relatively new class of tags targeting fluorogens (See works of Lavis, Gautier, Bruchez, Moerner). These tags should be highlighted since some of them have been shown to work with SMLM based techniques.

We carefully rechecked the literature and included the latest published work in the revised manuscript.

65-67 – what do the authors attribute the inferior signal from the monovalent tagged Nb to?

We thank the reviewer for this comment and agree that this detail was not properly stated in the original submission. We rephrased the sentence which now reads

“While BC2-Nb_{AF647} (NHS) is sufficient for wide-field microscopy (Fig. 1a, left panel, Supplementary Fig. 1a, b), dSTORM imaging of BC2-tagged proteins revealed a rather low staining efficiency resulting in inferior structural labeling coverage (Fig. 1b, left panel).” As stated above, our BLI assay demonstrates that the inferior signal of the monovalent BC2-Nb is due to a higher off-rate.

80-85 – While the authors report signal to noise here, the critical parameter to be reported when comparing labeling strategies is photon numbers/ localization and/ or the localization precision. Additionally, the SNR is a function of the background as well. The authors must report the background photons/ pixel in both the cases.

We now added a more detailed description of the analysis of **Fig 1a** and **Supplementary Fig. 1e** (wide-field images) including the requested parameters to the figure caption and the Methods section. For quantification of the obtained staining at a dSTORM level, we now added new experiments to the revised manuscript:

1) We assessed the completeness of labeling of the BC2-tag/bivBC2-Nb system by using the oligomeric bacterial protein FtnA as a protein standard with the defined number of 24 subunits. Our results show that our detection system has a labeling efficiency of about 61% which competes with the efficiency of the covalent SNAP-tag system of 64% in our comparative studies (matching previous studies of {Finan, 2015 #908; Szymborska, 2013 #915}). This is superior to efficiencies achievable with photoactivatable FPs. We added these results and corresponding references in the **main text, Fig 1c, new Supplementary Fig. 3 and Methods.**

2) We compared the background of unspecific staining by dSTORM imaging of various HeLa cells. To distinguish normal unspecific staining background from additional β -catenin staining, we compared HeLa cells (not expressing any GFP epitope) stained with a GFP-targeting nanobody to HeLa cells stained with bivBC2-Nb_{AF647}. Further, we measured HeLa cells transiently expressing the non-structural autophagosomal marker protein LC3B fused to the BC2-tag, which is – in absence of autophagy - homogeneously distributed throughout the cell. By analyzing the SRM data by DBSCAN clustering, we measured a slightly increased level for the bivBC2-Nb compared to the unspecific nanobody labeling density for the GFP-Nb. However, this level is considerably lower compared to the bivBC2-Nb staining of BC2_TLC3B expressing cells (updated **main text**, **Fig. 1c**, new **Supplementary Fig. 2a** and **Methods**).

Fig 2 – For each cell image shown, it must be mentioned if they are live/ fixed cells. Same is true about the image display. The authors must mention where ever there is a reconstruction (and outline the reconstruction type: scatter, histogram, Gaussian etc.)

All images except for images shown in **new Fig. 5** and new **Supplementary Fig. 12** are derived from chemically fixed cells using the same fixation protocol (**Methods**). For the revised manuscript, we included a clear fixed/live cell description in the **main text** and in all **figure captions**. Exact details on how the images are reconstructed are described in the **Methods** section. We discussed whether to place a shortened description of the reconstruction details into the figure captions. In the end we decided that we prefer to have a complete description including all processing steps together and thus did not change the paragraph of the Methods of our first submission during the revision. This original, unchanged version reads:

“For each image reconstruction, 10,000 – 20,000 imaging frames with an exposure time of 70 ms were recorded at a pixel size of 129 nm. The camera, microscope and AOTF were controlled by μ Manager software¹³ on a PC workstation. Single-molecule localizations were extracted from the movies with the open-source software Rapidstorm 3.2.¹⁸ Drift correction was performed by custom written Python 2.7 algorithms that extract and correct for fluorescent bead tracks. Nearest Neighbour Analysis (NeNA) as described in¹⁹ was done with the open-

source software *Lama*²⁰ on a section of the image that contained no fiducial markers. Localizations appearing within the radius of the NeNA value on several frames were grouped into one localization using the Kalman tracking filter in Rapidstorm 3.2. Final images were reconstructed at a pixel size of 10 nm. For visualization, a Gaussian blur filter was applied in the ImageJ software using NeNA as the sigma value.”

In the revised version we added a short linking sentence to all captions for dSTORM imaging which reads: “Image reconstruction details are given in the **Methods** section”.

104-106 – Is the phenotype here just smaller cells? Is this a live/ fixed cell comparison? What reasons do the authors attribute for this difference between GFP vs pa-mCherry labeling? Is there literature supporting this difference between two FPs? What if this is an artifact of fixation (see a recent paper from the Tijan lab where they discuss this)

As for the description of Fig. 2, we acknowledge that we did not present our data properly. All images were acquired from chemically fixed cells and our findings strongly indicate that FP-tagged vimentin (regardless which FP - GFP or PAmCherry - is used or whether the FP is placed on the N- or the C-terminus) is not fully incorporated into the endogenous vimentin network which results in the observed phenotypes. We included the notion on the differences in FP self-oligomerization of jelly fish FPs (e.g. GFP) and red coral FPs (e.g. Cherry) and referenced oligomerization artifacts studies for different FP classes {Wang, 2014 #829; Cranfill, 2016 #902}. The mentioned artifact described in the Tijan paper was measured for very fast protein dynamics, which interfered with the slower fixation kinetics and led to many proteins fixed in solution. In our view, this doesn't apply to our constructs which are built into rather stable protein structures. Furthermore, all the analyzed cells were subjected to identical fixation protocols. We also did not find any evidence of fixation-induced artifacts when imaging non-structural proteins added during the revision at a dSTORM resolution (updated **main text**, **Fig. 3**, **Supplementary Fig. 11** and **Methods**).

124-133 – Chromophore maturation times must be on the same order for PAmCherry and GFP. The authors must definitely perform a photoactivation light dose dependent experiment and infer the difference between GFP and pa-mCherry. This section is full of “likely” causes for experimental observations. The authors must provide some experimental evidence for some observations in the least.

All images were derived from chemically fixed cells after 24 h of transfection with the corresponding FP-fusion which ensure a complete expression and maturation of the corresponding FP. We acknowledge that our wording in this section was misleading. In the revised version, we rephrased this section, highlight the experimental evidence and also give appropriate references for the different behavior of the FP tags. Regarding *...photoactivation light dose dependent experiment and infer the difference between GFP and pa-mCherry...* we are sorry, but we don't fully understand the comment, as GFP is a “normal” FP and can't be photoactivated. The GBP-Nb_{AF647} recognizes a 3D epitope located at the beta-barrel structure of GFP and thus also recognizes the immature chromophore.

Also, independent of the FP used, our general FP-photoactivation procedure (here used for PA-mCherry) is to keep the spots per frame stable until all FPs are read out by ramping up the UV intensity. We included these details into our **Methods** section.

One big gap in the understanding about the delivery of these tags that the authors speculate about but do not clearly explain the delivery of Alexa647 labeled protein in the cells. It is known that Alexa647 is a membrane impermeable dye. It is possible that the Nb shields the dye charge by encapsulating/ forming some kind of secondary structure around the dye. One experiment that could be performed here is tagging dyes with different charges/ backbones (Cyanines vs Rhodamines, for example). The uptake for dyes with different charges/ backbones will help us further our understanding of the mechanism behind the uptake. This will also make the technology more general and widely used.

In preliminary experiments we have tested the uptake of bivBC2-Nb conjugated with different dyes such as ATTO488 (rhodamine based), CF568 (rhodamine based), CF647 (modified

cyanine) and AF647 (sulfonated cyanine). We observed staining of BC2-tagged structures with bivBC2-Nb conjugated to each of the dyes (see panel of live-cell staining below). From that it can be concluded that the chemical nature of the dye has no influence on uptake.

AO: ATTO488

Nevertheless, since our results exploring the exact uptake mechanism are still premature and we cannot provide a satisfactory explanation to the readership, we decided to remove the paragraph describing the uptake of the nanobody in the revised manuscript. We thus decided to focus on establishing a more robust and uptake-independent live-cell dSTORM protocol for the bivBC2-tag system. Here, using a commercially available lipid-based protein transduction reagent, we demonstrate that also all dye classes can be used (new **Supplementary Fig. 13b** as well as the protocol in the **Methods** section). We also updated **Fig. 5** which now shows that the BC2-tag/bivBC2-Nb system can also be used for live-cell dSTORM imaging and tracking studies.

Minor comments:

Please read the SI carefully before submission. There are several small things that have been overlooked. This is another aspect that should be improved drastically for submission to any journal. Noting only 2 examples here:

SI Fig 1 – The authors have confused between k_{off} and K_d in the table and legend for figure 1 in SI. Please write k_{off} in the table.

SI line 104 – Peptide Dissolved not solved

It will be good to provide a table comparing properties such as labeling concentrations, incubation times, # of washes for the Nb tag and conventional tags.

We carefully went through the SI as suggested and rephrased all comments raised by the reviewer accordingly. As the staining procedures were kept constant for all experiments and are described in detail in the **Methods**, we would like to refrain from including an additional table as proposed by the reviewer.

Reviewer 2

1) The primary impact/novelty of the manuscript is an improvement in labeling affinity and completeness with a structurally minimal tag. In this context, the heavy reliance on comparatively abundant structural proteins, the traditional low-hanging fruit of super-resolution microscopy, is not appropriate. This is puzzling given that the authors present data on non-structural proteins, but relegate it to supplementary figures with minimal analysis. A comparative attempt to colocalize two proteins known to co-complex in low copy number would be a useful addition to the paper.

In our 1st submission we were convinced that our data obtained for the components of the cytoskeleton needs to be shown with priority as structural proteins are sensitive to genetic tagging and easily show possible dysfunction or mislocalization caused by the tag. Thus, we focused on this aspect and demonstrated that BC2-tagged components are functionally incorporated into various endogenous structures.

We nevertheless fully agree with the reviewer that we neglected other important aspects when testing appropriate tags for SMLM and now saw to them and updated those with more detailed results during the revision. In the revised manuscript we now include dSTORM images of the BC2-tagged non-structural proteins LC3B and GFP-GPI which we present in the updated **Fig. 3d and Fig. 3e** and quantified the effect of rapamycin treatment on LC3B clustering on autophagosomes by DBSCAN clustering analysis (**Fig 3e** and new **Supplementary Fig. 11**): Notably, our findings regarding the size of autophagosomes are in perfect agreement with what is described in the literature {Jin, 2014 #916; Mizushima, 2002 #907}.

To quantitatively benchmark our labeling system, we further constructed a BC2-tagged version of the bacterial FtnA-24mer, a homo-oligomeric standard recently described by {Finan, 2015 #908}. Comparing the stainings of the FtnA-24mers for the SNAP-tag and bivBC2-Nb systems, our results show that our labeling approach competes with the efficiency of the covalent SNAP-tag system and is superior to efficiencies reached by photoactivatable FPs. We added these results as **new Fig. 1c** and **Supplementary Fig. 3**.

2) While the apparent uptake of nanobody into live cells for labeling purposes is extraordinary, treatment of this phenomenon should not be limited to images of immortalized cancer cell lines. Minimally there should be a quantitative treatment of the kinetics of this effect and this should be replicated in pombe; ideally I would like to see this on primary cells or tissue.

Initially we have demonstrated the uptake of the bivBC2-Nb for various cancer cell lines. However, our results exploring the exact uptake mechanism are still premature and we cannot provide a satisfactory explanation to the readership. Thus, we decided to remove the paragraph describing the uptake of the nanobody in the revision as it is not the main focus of this manuscript. Instead we established a more robust and uptake-independent live-cell dSTORM protocol for the bivBC2-tag system to demonstrate that the nanobody is functional in living cells. Using a commercially available lipid-based protein transduction reagent, we now demonstrate that the BC2-tag/bivBC2-Nb system can also be used for live-cell dSTORM imaging and tracking studies (new **Fig.5** and new **Supplementary Fig. 13**). We agree that expanding this approach to primary cells would be highly interesting. However, the generation of such genetically modified cell systems carrying BC2-tagged antigens would exceed our current possibilities.

3) It appears that every BC2 construct used was constructed as a C-terminal fusion. This should be tested in other configurations; if C-terminal structure is required for nanobody recognition, this should be discussed in detail.

It is not correct as author suggests in the summary that the native alpha-tubulin is unaffected by C-terminal additions. C-terminal post-translational modifications abound on tubulin, including multiple carboxy terminal residue cleavages. I find the assertion dubious for the other listed proteins as well in the absence of appropriate references. While a short peptide would be expected to have a less disruptive impact than a full fluorescent protein, the option to change sites can be critical to experiments and the feasibility of such a substitution should be examined in detail.

In our current manuscript different orientations of the BC2-tag are shown indicated by the given name (e.g. vimentin_{BC2T} or _{BC2T}lamin). The generation of the expression constructs is described in the Methods section. We agree with the concerns raised by the reviewer and acknowledge that like for any other tagging system the orientation of the tag and its influence on the corresponding protein has to be critically evaluated. Thus we add a new section in the main text for a more detailed description and tested whether orientation of the BC2-tag affects the incorporation of recombinant proteins into endogenous structures. Thus, we transiently expressed cDNAs of mouse *TUBA1B*, human *LMNB1* or *ACTB* either comprising the BC2-tag on the N- or the C-terminus in different cell lines followed by detection with bivBC2-Nb_{AF647}. In this context we show for tubulin alpha-1B, that C-terminal addition of the BC2-tag yielded more distinct microtubule structures compared to the N-terminally tagged version (new **Supplementary Fig. 8**). When we compared our findings with corresponding GFP-tagged tubulin we find that - like for the BC2-tag - expression of a C-terminal fusion of GFP results in more distinct tubulin filaments in HeLa, U2OS and COS-7 cells. Below please find a panel of representative images of GFP-tagged tubulin, which will not be included in the revised manuscript.

Minor comments:

1) N values for testing categories of filament width are not apparent. Statistical sections would benefit from editing for clarity.

In the revised manuscript we included N values for all figures with statistical analysis in the figures captions.

2) The methods section is otherwise exemplary, and demonstrates a commitment to reproducibility and broad adoption of new method. This level of detail is not common and I thank the authors for their diligence.

We thank the reviewer for this valuable appreciation.

3) The antigenic peptide tag is not particularly novel, but important to overall paper; in this respect it needs better characterization and discussion.

In the revised manuscript we add novel data analysis to characterize the BC2-tag more appropriate (please see other comments).

Reviewer 3

1) The new method is based on a nanobody, BC2-Nb, which binds to a peptide tag. BC2-Nb was originally developed as a nanobody against β -catenin that is endogenously expressing in various cells. Hence, the effect of endogenous β -catenin should be discussed in the manuscript.

Expression levels of proteins shown by dSTORM in this manuscript, i.e. actin, vimentin, and tubulin, are known to be high and therefore endogenous β -catenin might be negligible in these cases, but the expression level of most of the protein of interest is much lower and this reviewer was wondering if the effect of endogenous β -catenin becomes obvious.

We compared the background of unspecific staining by dSTORM imaging of various HeLa cells. To distinguish normal unspecific staining background from additional β -catenin staining, we compared HeLa cells (not expressing any GFP epitope) stained with a GFP-targeting nanobody to HeLa cells stained with bivBC2-Nb_{AF647}. Further, we performed bivBC2-Nb staining in HeLa cells transiently expressing the non-structural autophagosomal marker protein LC3B fused to the BC2-tag, which is – in the absence of autophagy - homogeneously distributed throughout the cytoplasm. By analyzing the SRM data by DBSCAN clustering, we measured a slightly increased level for the bivBC2-Nb compared to the unspecific labeling for the GFP-Nb. However, this level is considerably lower compared to the bivBC2-Nb staining of BC2_TLC3B expressing cells (**new Fig. 1c, new Supplementary Fig. 2a**, updated **main text** and **Methods**).

In a 2nd experimental scheme, we induced a strong enrichment of β -catenin by CHIR99021 treatment. While immunolabeling with a β -catenin-specific antibody showed a strong signal of the treated cells, dSTORM imaging of bivBC2-Nb_{AF647} staining revealed only a negligible increase in AF647 localizations. Moreover, in CHIR-treated HeLa cells transiently expressing vimentin_{BC2_T}, the nanobody signal was almost exclusively detectable at vimentin fibers. From this we concluded that even if present at high levels, the BC2 epitope of β -catenin has a minor impact on staining of ectopically introduced antigens (**new Supplemental Fig. 2b**).

We updated **main text** and the **Methods** section and added the **new Supplemental Fig. 2**.

2) Labeling in living cells by BC2-Nb is interesting, and this reviewer was convinced that it worked for wide-field fluorescence imaging. However, apparently the background is rather high and most probably the labeling efficiency is not as good as the labeling in fixed cells. Therefore, this reviewer suspects if the labeling in living cells is not applicable for dSTORM imaging. If it is applicable, please simply show an example. If not, it should be clearly stated in the text and explain, otherwise audience might get confused that the method is applicable for dSTORM under living conditions.

To clarify this concerns we included two new experiments for live-cell dSTORM imaging and tracking studies using the BC2-tag/bivBC2-Nb system:

First, we performed time-lapse imaging of HeLa cells transiently expressing $_{BC2T}GFP-GPI$. After a fast recruitment of the nanobody to its membrane-located antigen (**new Supplementary Fig. 13**) live-cell single-particle tracking dSTORM imaging allowed us to trace the highly dynamic movements of the $_{BC2T}GFP-GPI$ molecules along the plasma membrane (**new Fig. 5a, Supplementary movies 1 - 4**).

For bivBC2-Nb staining of intracellular targets in living cells we adapted a lipid-based protein transfection protocol and introduced bivBC2-Nb conjugated to different dyes into our newly generated cell lines HeLa- $_{BC2T}ACTB$ and A549- $_{BC2T}ACTB$ (**new Supplementary Fig. 13b and c**). We then performed live-cell dSTORM imaging using bivBC2-Nb $_{ATTO655}$ staining which revealed the intracellular actin network of HeLa- $_{BC2T}ACTB$ cells (**new Fig. 5b**).

We also documented the differences in performance of the ATTO655 fluorophore, which is outcompeted by AF647 both in blinking statistics and brightness (**new Supplementary Fig. 14, Supplementary movies 5 and 6**). These data demonstrate that the bivBC2-Nb is functional within living cells where it retains its outstanding binding capacities allowing for live-cell dSTORM imaging.

We updated **main text** and the **Methods** section and added the **new Fig. 5, Supplementary Fig. 14, Supplementary movies 1-6**.

Reviewers' comments:

Reviewer #1 (Remarks to the Author):

The authors have done a wonderful job of addressing all the comments raised by the reviewers. Addition of new results that improve the transparency as well as application data on the method, are really appreciated! These quantifications and clear comparisons will improve the adaptation of the technology by the scientific community.

Three comments that this reviewer has is about the live cell imaging data from SI. These comments should be addressed to clarify the live-cell imaging application and its limitations.

Fig 13a: Is the signal at 24 mins saturated for the given concentration of the probe? This is important information to assess the labeling saturation time; often required for quantitative biological imaging. It will be useful to have the time that it takes to saturate, say all actin in live cells at a given probe concentration. Put differently, if the authors considered a region of interest in the image, how does the fluorescence intensity change in the region with time? Does that curve saturate in 24 mins?

Fig 13b and c: There are fluorescent puncta in images where dyes Alexa647 and CF568 have been used. Are the probes getting sequestered in some cellular compartments? These fluorescent puncta are not so visible in the Atto655 case. So, maybe there is some dye dependent uptake by an organelle. Do these puncta disappear at a lower concentration of the probe? The authors should comment on this.

SI on Live cell imaging:

"HeLa Kyoto transiently expressing BC2TGFP-GPI, HeLa-BC2TACTB, or A549-BC2TACTB cells were plated at ~5000 cells per well of a µclear 96-well plate (Greiner Bio One, cat. # 655090) and cultivated at standard conditions. Next day, time-lapse imaging was performed in a humidified chamber (37 °C, 5 % CO₂) of a MetaXpress Micro XL system (Molecular Devices) at 40 x magnification. Time-lapse imaging with 4 - 5 min intervals was started immediately upon addition of bivBC2-NbAF647 (1 µg/ml)."

Was this imaging performed without any washes? If yes, this should be highlighted as this can be very useful to the live-cell imaging community. In case washing is used, it must be stated.

"For live-cell staining of HeLa-BC2TACTB and A549-BC2TACTB upon protein transduction of nanobodies, cells were placed in DMEMgfp-2 medium two hours after addition of transduction mix and imaged in hourly intervals."

Please explain the contents of this transduction mix. Additionally, was there no washing required in this case either? I do see these details in the D-STORM live cell imaging section but they must be presented for the live cell imaging as well.

Reviewer #2 (Remarks to the Author):

The authors have responded comprehensively to some, but not all, of my concerns.

In regards to labeling efficiency of non-structural proteins (point 1), I suggested the colocalization experiment because it would be easier to quantify label completeness in the presence of an independent label than in the case where you have many (but still an indeterminate number) of proteins with a single label. My concern about demonstrating label completeness remains.

In regards to the apparent uptake of nanobody into live cells (point 2), this was a significant finding that I am disappointed to see removed; the removal does in a way address the problem of having insufficient characterization of the cell uptake phenomenon, but it also significantly detracts from the novelty of this manuscript, as this appeared a considerable advance.

In regards to the C-terminal fusion comment (point 3), the authors provided a substantive and complete response, thank you.

Finally, in regards to minor point 3, the background due to endogenous beta-catenin appears to be substantial. This should be repeated with LC3B before concluding that the impact will be negligible.

Reviewer #3 (Remarks to the Author):

The authors addressed to all major criticisms from this reviewer properly, which improved the manuscript substantially. The reviewer recommends publishing this study.

Reviewer #1 (Remarks to the Author):

The authors have done a wonderful job of addressing all the comments raised by the reviewers. Addition of new results that improve the transparency as well as application data on the method, are really appreciated! These quantifications and clear comparisons will improve the adaptation of the technology by the scientific community.

We thank the reviewer for his/her valuable appreciation!

Three comments that this reviewer has is about the live cell imaging data from SI. These comments should be addressed to clarify the live-cell imaging application and its limitations.

Fig 13a: Is the signal at 24 mins saturated for the given concentration of the probe? This is important information to assess the labeling saturation time; often required for quantitative biological imaging. It will be useful to have the time that it takes to saturate, say all actin in live cells at a given probe concentration. Put differently, if the authors considered a region of interest in the image, how does the fluorescence intensity change in the region with time? Does that curve saturate in 24 mins?

We thank the reviewer for this suggestion! In the revised manuscript we included an additional data set summarized in a new supplementary figure (new **Suppl. Fig 13b**), where we determined the fluorescence intensities derived from the nanobody labeling at indicated regions of interest (ROI) over time. Quantification of the nanobody signal shows a saturation of the fluorescence intensity within 20 - 30 min.

Fig 13b and c: There are fluorescent puncta in images where dyes Alexa647 and CF568 have been used. Are the probes getting sequestered in some cellular compartments? These fluorescent puncta are not so visible in the Atto655 case. So, maybe there is some dye dependent uptake by an organelle. Do these puncta disappear at a lower concentration of the probe? The authors should comment on this.

To deliver the dye coupled BC2-Nb we used Pro-DeliverIN (OZ Biosciences), which is a lipid-based transfection reagent. According to the manufacturer, the protein is captured through non covalent electrostatic and hydrophobic interactions and lipid-protein aggregates are internalized followed by the cytoplasmic release of the protein. Unfortunately no detailed information is given on the precise uptake mechanism. However, similar to our observation all exemplary data provided by the manufacturer show fluorescent aggregates or puncta. Thus we assume that the

observed puncta can be traced back to the chemical transfection treatment and do not originate from sequestration of the nanobody to cellular compartments or organelles. In preparation of this manuscript we tested different concentrations of dye-labeled bivBC2-Nb and transfection reagent. In all cases we observed the formation of fluorescent aggregates. Representative for all dyes used, here we show the uptake and antigen binding for bivBC2-Nb_{AF647} in HeLa-_{BC2T}ACTB cells over time.

Arrows indicate fluorescent complexes which disappear once the nanobody is released and recruited to its antigen (_{BC2T}actin) within living cells. Since a similar image is already included in the revised manuscript (now **Supplementary Fig. 14a**) we refrained from showing this time lapse analysis in the second revision.

SI on Live cell imaging:

“HeLa Kyoto transiently expressing _{BC2T}GFP-GPI, HeLa-_{BC2T}ACTB, or A549-_{BC2T}ACTB cells were plated at ~5000 cells per well of a μ clear 96-well plate (Greiner Bio One, cat. # 655090) and cultivated at standard conditions. Next day, time-lapse imaging was performed in a humidified chamber (37°C, 5 % CO₂) of a MetaXpress Micro XL system (Molecular Devices) at 40 x magnification. Time-lapse imaging with 4 - 5 min intervals was started immediately upon addition of bivBC2-Nb_{AF647} (1 μ g/ml).”

Was this imaging performed without any washes? If yes, this should be highlighted as this can be very useful to the live-cell imaging community. In case washing is used, it must be stated.

We revised the methods section accordingly which now reads:

“For live-cell staining of _{BC2T}GFP-GPI, culture medium was replaced without washing by live-cell visualization medium DMEMgfp-2 (Evrogen, cat. # MC102) supplemented with 10 % FCS, 2 mM

L-glutamine and 1 µg/ml bivBC2-Nb_{AF647}. Time-lapse imaging with 4 - 5 min intervals was started immediately upon medium replacement.

“For live-cell staining of HeLa-BC2TACTB and A549-BC2TACTB upon protein transduction of nanobodies, cells were placed in DMEMgfp-2 medium two hours after addition of transduction mix and imaged in hourly intervals.”

Please explain the contents of this transduction mix. Additionally, was there no washing required in this case either? I do see these details in the D-STORM live cell imaging section but they must be presented for the live cell imaging as well.

We revised the methods section accordingly which now reads:

“For live-cell staining of HeLa-_{BC2T}ACTB and A549-_{BC2T}ACTB upon protein transduction of nanobodies, cells were washed once with and placed in DMEMgfp-2 medium two hours after addition of transduction mix (see protein transduction section above) and imaged in hourly intervals.”

Reviewer #2 (Remarks to the Author):

The authors have responded comprehensively to some, but not all, of my concerns.

In regards to labeling efficiency of non-structural proteins (point 1), I suggested the colocalization experiment because it would be easier to quantify label completeness in the presence of an independent label than in the case where you have many (but still an indeterminate number) of proteins with a single label. My concern about demonstrating label completeness remains.

In the original manuscript we have quantified the labeling completeness of BC2-tagged vimentin fibers in comparison to three alternative stainings and could demonstrate extraordinary high labeling coverage even for the thinnest fibers (**Fig. 2, Suppl. Fig. 4, 6 and 7**) which, combined with our data on high localization precision and small label displacements, nicely revealed the potential of our new labeling system. Furthermore, we demonstrate this high performance for other structural proteins and quantified the labeling completeness it for actin and tubulin (**Fig. 3 and Suppl. Fig. 9**).

After the request of this reviewer for more data on labeling completeness, we then - during the 1st revision phase - decided on the quantitative measure of counting the individual subunits of a defined homo-oligomer, as this experiment, in comparison to a co-localization experiment, has the additional advantage to directly quantify the overall labeling efficiency of our system.

We chose this experiment as we believe that this data yields the strongest proof of the potential of our new STORM labeling system. It nicely avoids several possible caveats, e.g. incomplete labeling of a reference label, native protein background for transient transfections, registration errors (two channels, chromatic aberration, chromatically shifted focal plane etc. which strongly affects the nanometer-precise STORM data).

Indeed, we were able to show that our BC2T/bivBC2-Nb system can compete with covalent SNAP-tag labeling (**Fig. 1, Suppl. Fig. 3**) and at the same time reduces the size of the genetic tag (SNAP-tag, 182 aa) drastically to a short peptide tag (12 aa).

Reevaluating our strategy during this 2nd revision, we concluded that an extra co-localization experiment cannot provide additional crucial information and thus decided for no further experiments. We apologize that we didn't follow the reviewer's advice in the first place and instead designed and performed another study but kindly ask the reviewer for his/her appreciation that the "FtnA oligomer data" nicely quantifies the labeling efficiency of our BC2T/bivBC2-Nb system.

In regards to the apparent uptake of nanobody into live cells (point 2), this was a significant finding that I am disappointed to see removed; the removal does in a way address the problem of having insufficient characterization of the cell uptake phenomenon, but it also significantly detracts from the novelty of this manuscript, as this appeared a considerable advance.

We strongly disagree that there is only little novelty left in our manuscript. Here we show for the first time:

- 1) A conceptual strategy to generate the first peptide-specific nanobody for genuine STORM imaging by introduction of a bivalent nanobody format in combination with site-directed, 1:1 dye conjugation.
- 2) The detailed quantification of the BC2T/bivBC2-Nb labeling system for
 - Labeling efficiency, localization precision (NeNA) and resolution (FIRE) and apparent filament width for all major structural proteins including vimentin, actin and tubulin
 - Labeling efficiency for FtnA homo-oligomers in comparison to the covalent SNAP tag system
 - Determination of background derived from native beta-catenin in HeLa cell stainings for wildtype, LC3B and vimentin as well as under the strong induction of endogenous beta-catenin.
 - Assessment of live cell stainings by temporal imaging sequences.
- 3) The impact of the labeling on native biological phenotypes for
 - vimentin in comparison to three other labeling approaches
 - tubulin and vimentin concerning C-terminal and N-terminal tagging in several cell lines
 - nuclear yeast protein cbp1 by phenotype assessment
- 4) The application of the system for
 - Several structural proteins (vimentin, tubulin, lamin, actin)
 - Several non-structural molecules (LC3B, GPI, cbp1 in yeast)
- 5) The utility of the system for stable cell lines
 - in HeLa and A549 cells by CRISPR/Cas9 technology
 - in *S. pombe* yeast
- 6) The utility and quantification of the system for live cell approaches
 - Single-particle tracking STORM of membrane proteins
 - Live cell structural STORM of actin

We strongly believe that this manuscript provides substantial and plentiful data and thus easily overcomes the loss of the retracted live cell uptake data which we are currently evaluating in more detail due to more complex findings. We nevertheless would like to apologize for this, as we initially included these exciting findings and now must disappoint this reviewer in this regard.

In regards to the C-terminal fusion comment (point 3), the authors provided a substantive and complete response, thank you.

We thank the reviewer for his/her valuable appreciation!

Finally, in regards to minor point 3, the background due to endogenous beta-catenin appears to be substantial. This should be repeated with LC3B before concluding that the impact will be negligible.

We are very sorry, but we do not understand this comment. In the current manuscript we already quantified BC2-tagged LC3B, which shows a significant specific staining above background. The data can be found in **Figure 1 c** and corresponding images showing the DBScan analysis can be seen in **Suppl. Fig. 2a**. Next to this comparison of wildtype HeLa cells and LC3B, we further quantified a possible native beta-catenin background by staining cellular structures upon induction of endogenous beta-catenin. This is shown in all detail in **Suppl. Fig. 2b**. We strongly believe that these analyses leave no open questions regarding the impact of native beta-catenin background and clearly show that our BC2T/bivBC2-Nb system is perfectly capable to visualize even low abundant and non-structural proteins such as LC3B.

Reviewer #3 (Remarks to the Author):

The authors addressed to all major criticisms from this reviewer properly, which improved the manuscript substantially. The reviewer recommends publishing this study.

We thank the reviewer for this valuable appreciation!